# Local and remote climatic drivers of extreme summer sea surface temperatures in the Arabian Gulf

Zouhair Lachkar<sup>1</sup>, Olivier Pauluis<sup>1,2</sup>, Francesco Paparella<sup>1</sup>, Basit Khan<sup>1</sup>, and John A. Burt<sup>1,3</sup>

Correspondence: Zouhair Lachkar (zouhair.lachkar@nyu.edu)

# Abstract.

The Arabian Gulf (also known as the Persian Gulf; hereafter simply the Gulf) is a shallow, subtropical, semi-enclosed sea that experiences the highest average summer sea surface temperatures (SSTs) in the global ocean. While local marine organisms have adapted to these extreme conditions, interannual temperature fluctuations result in occasional marine heatwaves (MHWs) that trigger mass coral bleaching and other ecological impacts. However, the climatic drivers of this variability remain poorly understood. In this study, we investigate the sources and mechanisms behind extreme summer SSTs in the Gulf. Analyzing a regional eddy-resolving ocean hindcast simulation combined with ERA5 reanalysis data, we find that extreme summer surface temperatures are tightly linked to lower-than-normal surface pressure over the Arabian Peninsula and higher-than-normal surface pressure over Iran and Pakistan. These pressure anomalies are typically associated with stronger monsoonal winds in the western Arabian Sea and weakened local winds. Enhanced monsoon circulation leads to (i) increased evaporation over the Arabian Sea and greater moisture transport into the Gulf in the lower troposphere, trapping heat near the surface, and (ii) enhanced subsidence over the Arabian Peninsula and the Gulf in the upper troposphere, inducing further surface heat retention. Meanwhile, weakened local Shamal winds—particularly in the northern Gulf—reduce evaporative cooling and facilitate the accumulation of moist, warm air, amplifying local warming. These regional atmospheric changes are strongly modulated by large-scale climate variability modes, with El Niño-Southern Oscillation (ENSO) and the North Atlantic Oscillation (NAO) together explaining over 50% of the observed interannual SST variability in the Gulf. La Niña (El Niño) and the negative (positive) phase of the NAO both favor weaker (stronger) Shamal winds and warmer (cooler) SSTs during the peak summer months over the Gulf. Additionally, La Niña (El Niño) is associated with stronger (weaker) monsoon winds in the southern and western Arabian Sea, leading to stronger and more persistent warming (cooling) anomalies in the Gulf. The influence of these teleconnections is additive, with the warmest summers occurring when La Niña and negative NAO phases coincide. These findings have implications for the predictability of summer MHWs and associated ecosystem risks in the Gulf.

## 1 Introduction

The Gulf is a shallow, subtropical, semi-enclosed sea that experiences the highest average summer SSTs in the global ocean (Vaughan et al., 2019). Although local marine organisms have evolved to tolerate these extreme conditions, many species are

<sup>&</sup>lt;sup>1</sup>Mubadala Arabian Center for Climate and Environmental Sciences, New York University Abu Dhabi, Abu Dhabi, UAE

<sup>&</sup>lt;sup>2</sup>Courant Institute of Mathematical Sciences, New York University, New York, USA

<sup>&</sup>lt;sup>3</sup>Water Research Center, New York University Abu Dhabi, Abu Dhabi, UAE

believed to live near their upper thermal tolerance limits. As a result, MHWs driven by interannual temperature fluctuations can trigger coral bleaching and mass mortality events, even during summers when temperatures rise by only 2°C above the long-term average (Riegl et al., 2018; Burt et al., 2019). In addition to thermal stress, extreme temperatures increase the metabolic oxygen demand of ectothermic organisms like fish, which face significant physiological strain. These species must consume more oxygen under elevated temperatures while simultaneously being challenged by lower dissolved oxygen concentrations—a consequence of both warming-induced decline in oxygen solubility and enhanced water column stratification in the Gulf (Vaughan et al., 2021; Lachkar et al., 2022, 2024). The long-term physiological burden of coping with the Gulf's naturally extreme temperature has also been linked to reduced body size and productivity in Gulf fish populations, thereby negatively impacting fisheries—a sector of critical socio-economic importance (Ben-Hasan et al., 2024; Johansen et al., 2024). Finally, extreme summer temperatures in the Gulf may also affect regional extreme precipitation, given the Gulf's role as a key moisture source for the surrounding atmosphere (Pathak et al., 2025).

30

Several previous studies have investigated Gulf SST variability and explored the potential influence of large-scale climate oscillations on extreme temperatures in the Gulf. For example, Purkis and Riegl (2005) noted a similarity between the frequency of ENSO events and extreme Gulf SSTs, and speculated that ENSO might influence Gulf temperatures indirectly via its impact on the Indian Ocean Dipole (IOD). Al-Rashidi et al. (2009) suggested that El Niño could affect summer SSTs in the Gulf, linking the record-high SSTs of summer 1998, and the associated coral reef mortality (Riegl, 2003), to the El Niño event that developed in late 1997. Nandkeolyar et al. (2013) examined SST variability in several marginal seas of the Arabian Sea during 1985–2009, highlighting long-term warming trends while also proposing that ENSO and IOD modulate interannual SST variability. However, that study did not identify a consistent one-to-one correspondence between SST anomalies and these climate modes. More recently, Al Senafi (2022) applied an Empirical Orthogonal Function (EOF) analysis to identify dominant patterns of SST variability in the Gulf, attributing the first three principal components to the Atlantic Multidecadal Oscillation (AMO), ENSO, and IOD. Bordbar et al. (2024) analyzed Gulf SSTs between 2003 and 2021, and suggested that positive IOD events may favor elevated SSTs in the region. However, most previous studies relied on coarse-resolution datasets (e.g., Purkis and Riegl, 2005), were spatially or temporally limited (e.g., Al-Rashidi et al., 2009; Nandkeolyar et al., 2013; Bordbar et al., 2024), or analyzed multi-seasonal data without isolating the summer period, when extreme SSTs predominantly occur (e.g., Nandkeolyar et al., 2013; Al Senafi, 2022; Bordbar et al., 2024). Moreover, no study has yet systematically investigated the mechanisms through which large-scale climate modes influence Gulf SSTs, nor quantified the relative contributions and interactions among these modes. As a result, the drivers of summer SST extremes and the mechanisms through which largescale climate modes influence Gulf SSTs remain poorly understood.

This study aims to fill these gaps by identifying the local and remote climatic drivers of extreme summer SSTs in the Gulf. Specifically, we address three key questions: (i) what are the dominant drivers of Gulf SSTs during summer?, (ii) what atmospheric conditions accompany extreme summer SSTs in the Gulf?, and (iii) to what extent, and through what mechanisms, are these conditions influenced by large-scale climate teleconnections?

To answer these questions, we analyze an eddy-resolving hindcast simulation of the Gulf along with ERA5 reanalysis data over the period 1980–2018. Our results show that extreme summer SSTs in the Gulf are associated with lower surface pressure

over the Arabian Peninsula and higher pressure over Iran and Pakistan in the lower troposphere. This pressure configuration weakens the predominantly northwesterly winds over Iraq, most of the Arabian Peninsula, and the Gulf, known as Shamal, and strengthens monsoon winds over the western and southern Arabian Sea. These changes enhance the transport of atmospheric moisture into the Gulf and increase upper-tropospheric subsidence, both of which contribute to heat trapping near the surface and elevated SSTs. Additionally, the reduction in evaporative cooling due to weaker local winds further amplifies the warming. We show that these atmospheric conditions are strongly favored during La Niña and negative NAO summers, with the warmest summers occurring when both modes co-occur. In contrast, limited warming is observed when the two modes are in opposite phases. Our results also reveal that major climate teleconnections influence Gulf temperatures more strongly, and often in ways that contrast with their impact on surface temperatures in the Arabian Sea.

## 2 Methods

75

We analyze the interannual variability of Gulf SSTs over the period 1980–2018 using two complementary data sources: (i) an eddy-resolving ocean hindcast simulation, which allows for quantification of the key drivers of SST variability at relatively high spatial resolution; and (ii) reanalysis data, which is used to investigate the link between local atmospheric changes over the Gulf and large-scale atmospheric circulation.

# 2.1 Hindcast Simulation

The circulation model is based on the Regional Ocean Modeling System (ROMS) (Shchepetkin and McWilliams, 2005). The model uses non-local K-profile parameterization (KPP) scheme for vertical mixing (Large et al., 1994). Advection is calculated using a rotated-split, third-order upstream-biased scheme following Marchesiello et al. (2009), which has been shown to reduce numerical dispersion while maintaining low diffusion. Covering the Indian Ocean from 31.5°S to 31°N and 30°E to 120°E, the model employs a horizontal resolution of 1/10 degree and 32 sigma-coordinate vertical layers, with enhanced resolution near the surface. Seafloor bathymetry is derived from the one arc-minute ETOPO1 dataset (Amante and Eakins, 2009). Open lateral boundary conditions are specified along the southern and eastern boundaries of the domain. For depth-averaged velocity and sea surface height, radiation boundary conditions following Flather (1976) and Chapman (1985), respectively, are applied. Barotropic tidal forcing is imposed at the open boundaries using eight tidal constituents (K2, S2, M2, N2, K1, P1, O1, O1) obtained from the TPXO8 global tidal model (Egbert and Erofeeva, 2002). Finally, river discharge from major rivers within the model domain, including the Shatt al-Arab in the northeastern Gulf, is imposed as point sources. Monthly climatological runoff from rivers discharging into the northern Indian Ocean, including the Arabian Sea and the Gulf, is incorporated using the global discharge dataset of Dai and Trenberth (2002). The hindcast simulation is forced by ECMWF ERA-Interim 6-hourly heat fluxes, air temperature, pressure, humidity, precipitation, and winds spanning January 1980 to December 2018. The initial and lateral boundary conditions for various parameters are derived from the ECMWF Ocean Reanalysis System 5 (ORAS5; Zuo et al., 2019) and World Ocean Atlas 2018 Garcia et al. (2019). Further details of the model setup are provided in Lachkar et al. (2024). The performance of the model in reproducing key aspects of the hydrography of the region was extensively evaluated in Lachkar et al. (2024). In general, the model captures the essential hydrographic features of the Gulf region, including the seasonal progression of temperature, salinity, and Gulf outflow, as well as the long-term trends in sea surface temperatures. A detailed description of the model evaluation is provided in Lachkar et al. (2024).

## 95 2.2 SST tendency equation

In order to understand the drivers of SST variability in the Gulf, we consider the SST tendency equation:

$$\frac{\partial T}{\partial t} = \overbrace{\nabla \cdot K \nabla T}^{\text{Diffusion}} - \overbrace{u \cdot \nabla_{\text{h}} T}^{\text{Horizontal advection}} - \underbrace{w \frac{\partial T}{\partial z}}^{\text{Vertical advection}} + \underbrace{\frac{Q_{\text{net}}}{Q_{\text{net}}}}_{Q_{\text{net}}Q_{\text{net}}Q_{\text{net}}}$$

where T is the SST, K is the diffusivity tensor, and where  $\nabla$  and  $\nabla_h$  are the 3-D and horizontal gradient operators, respectively. The symbols u and w denote the horizontal and vertical velocities, respectively, while  $\rho_w$ ,  $c_{p,w}$  and h are the water density, specific heat capacity and thickness of the model's surface layer.  $Q_{\text{net}}$  is the atmospheric net surface heat flux.

# 2.3 Formulation of Surface Heat Fluxes

The net heat flux,  $Q_{\text{net}}$ , at the ocean surface is the sum of four components: solar shortwave radiation  $(Q_{\text{sw}})$ , longwave radiation  $(Q_{\text{lw}})$ , latent heat flux  $(Q_{\text{lat}})$ , and sensible heat flux  $(Q_{\text{sen}})$ :

$$Q_{\text{net}} = Q_{\text{sw}} + Q_{\text{lw}} + Q_{\text{lat}} + Q_{\text{sen}} \tag{1}$$

The longwave radiation,  $Q_{lw}$ , is further decomposed into downward radiation ( $Q_{dlw}$ ) and upward radiation ( $Q_{ulw}$ ):

$$Q_{\rm lw} = Q_{\rm dlw} + Q_{\rm ulw} \tag{2}$$

While the solar radiation  $Q_{\text{sw}}$  and the downward longwave radiation  $Q_{\text{dlw}}$  are prescribed from the atmospheric reanalysis used to force the model, the outgoing longwave radiation  $Q_{\text{ulw}}$  is estimated from the SST using the Stefan–Boltzmann law:

$$Q_{\rm ulw} = \epsilon \sigma T^4 \tag{3}$$

where  $\epsilon$  and  $\sigma$  are the emissivity of the ocean surface and the Stefan–Boltzmann constant, respectively, and T is the SST expressed in Kelvin.

The latent and sensible heat fluxes are parameterized using bulk formulae following Fairall et al. (1996):

$$Q_{\text{lat}} = -\rho_a c_e L U \Delta Q, \quad \text{with } \Delta Q = (Q^S - Q^{\text{air}})$$
(4)

$$Q_{\rm sen} = \rho_a c_p c_s U \Delta T, \quad \text{with } \Delta T = (T^{\rm air} - SST)$$
 (5)

where  $\rho_a$  and  $c_p$  are the air density and the specific heat of air at constant pressure, respectively;  $c_e$  and  $c_s$  are the bulk transfer coefficients for latent and sensible heat; L is the latent heat of vaporization; U is the wind speed at 10 m;  $\Delta Q$  is the humidity gradient;  $Q^S$  and  $Q^{\rm air}$  are the saturated specific humidity at the sea surface and the specific humidity at 10 m, respectively;  $\Delta T$  is the air-sea temperature difference; and  $T^{\rm air}$  and SST are the surface air and sea temperatures.

To characterize the relationship between anomalies in atmospheric heat fluxes and SST, we analyze the statistical correlations between interannual anomalies of  $Q_{\text{net}}$  and anomalies in the SST tendency term  $\frac{d\text{SST}}{dt}$ . Additionally, to quantify the drivers of Gulf SST variability, we regress SST on the following heat flux components: solar shortwave radiation  $(Q_{\text{sw}})$ , downward longwave radiation  $(Q_{\text{dlw}})$ , latent heat flux  $(Q_{\text{lat}})$ , and sensible heat flux  $(Q_{\text{sen}})$ .

# 2.4 Reanalysis Data

120

130

135

ERA5 reanalysis data was retrieved from the Copernicus Climate Change Service (C3S) Climate Data Store (CDS) (Hersbach et al., 2023). Data are provided on a regular latitude–longitude grid with a spatial resolution of 0.25°. For the purpose of this study, the data were extracted for a domain encompassing the Arabian Peninsula, the Gulf, and the Arabian Sea, extending from the equator to 35°N and from 40°E to 80°E. The reanalysis data, used for composite and regression analyses, covers the same period as the hindcast simulation, that is, 1980 to 2018. For the Self-Organizing Map (SOM) analysis, the data was extended to the entire 1950–2024 period to ensure robust training of the neural network, as described in Section 2.5. In addition to SST, the ERA5 dataset includes the following meteorological variables: 2-metre air temperature; 10-metre wind speed and its zonal and meridional components; sea level pressure; evaporation rate; total column water vapor; 2-metre relative humidity; cloud cover; and, in the free atmosphere, geopotential height, specific humidity, and wind vectors at 850 hPa (lower troposphere) and 300 hPa (upper troposphere).

Finally, to assess the robustness of the key findings with respect to the choice of reanalysis product, we additionally use two independent atmospheric reanalysis datasets: the Japanese 55-year Reanalysis (JRA-55; Kobayashi et al., 2015) and the Modern-Era Retrospective Analysis for Research and Applications, Version 2 (MERRA-2; Gelaro et al., 2017). Data from these datasets were extracted over the same study period and include the following variables: sea level pressure, 10-m winds, 850 hPa winds and geopotential height, and total-column water vapor.

# 2.5 Climate Teleconnections

To explore the potential influence of large-scale natural climate variability on extreme temperatures in the Gulf, we consider four global and regional climate variability modes known to affect the climate of the Middle East and the Arabian Peninsula: ENSO, NAO, IOD, and the Indian Summer Monsoon (ISM). Other modes, such as the Arctic Oscillation, were also examined but showed very weak and statistically insignificant correlation with summer Gulf SST (Fig S1, Supplementary Information

(SI)), and were therefore excluded from further analysis. The intensity and phase of each mode are characterized using the following state-of-the-art indices:

- For ENSO, we use the Oceanic Niño Index (ONI), defined as the 3-month running mean of sea surface temperature anomalies in the Niño 3.4 region (5°N-5°S, 120°-170°W), based on NOAA's ERSST.v5 dataset (Huang et al., 2017). The anomalies are calculated relative to centered 30-year base periods, updated every 5 years.
- For NAO, we use the NAO Index, which is based on the difference in sea-level pressure between the Subtropical (Azores)
   High and the Subpolar Icelandic Low following Barnston and Livezey (1987). The index, standardized and smoothed using a 3-month running mean, is obtained from the NOAA Climate Prediction Center website.
  - The IOD is represented by the Dipole Mode Index (DMI), defined as the anomalous SST gradient between the west-ern equatorial Indian Ocean (50°E−70°E, 10°S−10°N) and the southeastern equatorial Indian Ocean (90°E−110°E, 10°S−0°N). DMI values are obtained from the NOAA Physical Sciences Laboratory (PSL) website.
- Finally, to characterize the strength of the Indian Summer Monsoon, we use the Webster-Yang Monsoon Index (WYMI), a dynamical index defined as the vertical shear of zonal wind between the upper and lower troposphere over the region extending from 0° to 20°N and from 40°E to 110°E (Webster and Yang, 1992). A stronger WYMI reflects enhanced low-level westerlies and upper-level easterlies—both manifestations of a vigorous ISM circulation.

# 2.6 Statistical Analysis

165

170

To extract the interannual variability in Gulf temperature and its drivers, the data were deseasonalized by removing the monthly climatological means from the original time series, and detrended by subtracting the long-term linear trends over the study period. Specifically, for a variable A, we considered the interannual anomaly A', defined as:

$$A' = A - A^{\text{clim}} - A^{\text{trend}} \tag{6}$$

where  $A^{\rm clim}$  and  $A^{\rm trend}$  represent the monthly climatology and the linear trend of the variable A, respectively. The analysis was further restricted to the summer months (July to September), when extreme temperatures are most likely to occur. We tested the normality of the data using the Shapiro–Wilk test (Table S1, SI) and by visually inspecting quantile–quantile (Q–Q) plots, which compare the data quantiles to the theoretical quantiles of a normal distribution (Fig. S2, SI). The results of the test, along with the visual inspection, indicate that all variables are either normally distributed or approximately normal. Given that the data do not substantially deviate from normality, we chose to use parametric methods to analyze statistical correlations. We computed local Pearson correlation coefficients between SST anomalies and anomalies in potential drivers at each grid point, as well as correlations between Gulf-mean SST anomalies and regional or global climate variability modes. The statistical significance of the correlations was assessed using a Student's t-test at a 95% confidence level. Because these modes are not independent and exhibit mutual correlations, we further conducted a partial regression analysis to isolate the unique relationship between each mode and Gulf SST anomalies while statistically controlling for the influence of the others. This approach involves regressing

the residuals of SST (after removing the effects of all other predictors) against the residuals of each individual climate mode (after removing the effects of the remaining modes).

To quantify the typical SST response to variations in surface heat fluxes, we perform a composite of SST anomalies corresponding to the difference between high (> +1 SD above the mean) and low (

Figure 1. Inter-annual variability of the Gulf summer SST. (A) Time evolution of Gulf-averaged summer SST between 1980 and 2018 (black curve). The grey shading indicates the spatiotemporal variability, represented by  $\pm$  1 standard deviation around the Gulf-averaged summer SST. The red and the blue curves show the absolute local maximum and minimum summer SSTs within the Gulf, respectively. (B-C) Mean summer (July-September) SST (B) and inter-annual standard deviation (C) over the study period (1980-2018). The Gulf basin, Sea of Oman, and Strait of Hormuz are indicated in panel B.

## 3 Results

210

## 3.1 Summer SST Variability in the Gulf and Its Drivers

The summer SST in the Gulf exhibits pronounced spatial and temporal variability. During the study period (1980–2018), average summer SSTs ranged from below 28°C to above 35°C across the Gulf (Fig. 1A). This variability reflects interannual fluctuations superimposed on a long-term warming trend, further modulated by spatial heterogeneity. On average, SSTs are generally higher in the southern part of the Gulf compared to the northern region (Fig. 1B). When spatial variations are averaged out, the Gulf-wide mean summer SST increased at a rate of approximately  $0.31^{\circ}$ C per decade over the study period (Fig. 1A). Superimposed on this long-term warming trend are substantial interannual anomalies, with the most extreme basin-scale positive and negative departures occurring in the summers of 1998 and 1991, respectively (Fig. 1A). The magnitude of these interannual fluctuations also displays strong spatial variability, with weaker anomalies near the Strait of Hormuz (SD  $\approx$  0.3°C) and stronger anomalies in the northern Gulf (up to SD  $\approx$  0.7°C; Fig. 1C).

A heat budget analysis of the surface layer over the study period reveals that anomalies in the SST tendency term  $\frac{dSST}{dt}$  primarily reflect the near-compensating effects of anomalies in atmospheric heat fluxes ( $r = 0.88^*$ ) and vertical transport processes (mixing and advection;  $r = -0.87^*$ ) (Fig. 2A). In contrast, anomalies in Gulf-integrated lateral heat transport—associated with heat exchange with the Sea of Oman—exhibit only a weak correlation with the SST tendency (r = 0.25; Fig. 2A). The strong anticorrelation between anomalies in atmospheric heat fluxes and vertical transport ( $r = -0.99^*$ ) further indicates that vertical transport largely acts as a response to surface forcing rather than an independent driver of SST variability (Fig. 2A). Overall, at the scale of the entire Gulf, variations in atmospheric heat fluxes emerge as the dominant control on SST variability. Spatially, this influence is strongest in the northern Gulf, where atmospheric heat fluxes account for 92% of the SST variance (Fig. 2B), and remains dominant in the southern Gulf, accounting for about 83% (Fig. 2C).

Warm summer SST anomalies are associated with positive anomalies in downward longwave radiation and, to a lesser extent, latent heat flux—particularly in the northern Gulf (Fig 3). However, high summer SSTs are generally accompanied by weaker-than-average sensible heat flux and incoming solar (shortwave) radiation (Fig. 3). The reduced sensible heat flux from the atmosphere to the ocean surface during warm summers results from the combined effect of weaker winds and a smaller air—sea temperature difference ( $\Delta T = T^{air}$  – SST; defined in Eq. 5), as air temperature does not increase as much as SST during these periods (Fig. 4A–B and Fig. S3, SI). The weaker-than-average shortwave radiation is likely due to increased atmospheric moisture content during warm SST summers, since changes in cloud cover are not statistically significant (Fig. 4D–E). The contrasting impact of atmospheric moisture on surface shortwave and longwave radiation fluxes is further evidenced by the strong anticorrelation between anomalies in downward longwave and shortwave radiation (Fig. S4, SI).

Finally, changes in evaporative cooling—which depend on both the air-sea humidity gradient ( $\Delta Q = Q^s - Q^{air}$ ; defined in Eq. 4) and wind speed—are governed by the sign and relative contribution of these two factors. The surface saturation specific humidity ( $Q^s$ ) and air specific humidity ( $Q^{air}$ ) increase with rising SST and air temperature, respectively. However, because SST anomalies tend to be larger than air temperature ( $T^{air}$ ) anomalies during warm Gulf SST summers, and because relative

Figure 2. Drivers of interannual variability of summer Gulf SST. (A) Interannual anomalies of the temperature tendency term  $\frac{dSST}{dt}$  (black), atmospheric heat fluxes (red), and vertical (blue) and horizontal (green) heat transport fluxes, integrated over the entire Gulf. Correlation coefficients (r) in the legend indicate the Pearson correlation between each heat flux and the SST tendency term. (B-C) Inter-annual summer SST anomaly (black) as a function of time, along with the contribution of atmospheric heat fluxes to the SST anomaly (red), in the northern (B) and southern (C) Gulf. Note that in (A) the tendency term and horizontal transport fluxes are plotted in °C per year, while the atmospheric heat fluxes and vertical transport fluxes are shown in °C per month, allowing all four terms to be displayed on the same scale. The contribution of heat fluxes in (B) and (C) is estimated from a linear regression of SST anomalies onto anomalies in the four heat flux components: solar shortwave radiation, downward longwave radiation, latent heat flux, and sensible heat flux.

humidity typically decreases with increasing air temperatures, the humidity gradient ( $\Delta Q$ ) increases under these conditions 240 (Fig. 4C).

Figure 3. Relationship between atmospheric heat fluxes and Gulf SST. (Left): Pearson correlation coefficients between SST anomalies and atmospheric heat flux anomalies. (Right): Composite SST anomalies corresponding to the difference between high (> +1 SD above the mean) and low (

Figure 4. Correlations between SST and meteorological variables over the Gulf. Correlation between SST and wind speed (A), air-sea temperature difference  $\Delta T = T^{air}$  - SST (B), air-sea humidity gradient  $\Delta Q = Q^S$  -  $Q^{air}$  (C), total column water vapor (D), cloud cover (E), and evaporation (F). Hatching indicates statistical significance at 95% confidence interval.  $T^{air}$  and  $Q^{air}$  refer to air temperature and specific humidity at 2 m.  $Q^S$  is the saturation specific humidity at sea surface.

Figure 5. Climatological atmospheric conditions over the Gulf region in summer. Climatological mean sea level pressure (in mb) and surface (10 m) winds (A), total column water vapor (in kg m $^{-2}$ ) (B), geopotential height (in m) and winds at 850 hpa (C), vertical velocities (in hpa day $^{-1}$ ; negative values indicate upward currents) at 850 hpa (D), geopotential height (in m) and winds at 300 hpa (E), and vertical velocities (in hpa day $^{-1}$ ; negative values indicate upward currents) at 300 hpa (F) during summer months.

In the southern Gulf and the Sea of Oman, where wind changes are weak or uncorrelated with SST anomalies, evaporation increases, leading to cooler SSTs (Fig. 4F). In contrast, in the northern Gulf, the pronounced weakening of the Shamal wind during warm SST summers offsets the modest increase in humidity gradient ( $\Delta Q$ ), resulting in reduced evaporation and, consequently, diminished evaporative cooling (Fig. 4F and Fig. S3, SI).

In summary, higher summer SSTs in the Gulf appear to be associated with enhanced downward longwave radiation, and—locally in the northern Gulf—by reduced evaporative cooling. In the following section, we investigate the large-scale atmospheric circulation patterns that modulate these surface heat fluxes and contribute to the occurrence of extreme summer SSTs in the Gulf.

# 3.2 Influence of Large-Scale Atmospheric Circulation on Extreme Summer SSTs in the Gulf

260

The mean atmospheric circulation near the surface and in the lower troposphere during a typical summer is characterized by a high-pressure system over East Africa, the western Arabian Peninsula, and the southern Arabian Sea, and a low-pressure system over the eastern Arabian Peninsula, the northern Arabian Sea, southern Iran, and Pakistan (Fig. 5). This configuration favors the Shamal winds, while the pressure gradient over the Arabian Sea drives strong southwesterly monsoon winds across the Arabian Sea (Fig. 5A–C). While the Shamal winds transport dry air from Iraq and the northern Arabian Peninsula into the Gulf, the monsoon-associated southwesterly winds bring moisture-rich air to the southeastern Arabian Peninsula and the Gulf (Fig. 5B). Finally, the low-pressure system over the Arabian Peninsula is associated with ascending air in the lower troposphere, while the upper troposphere is dominated by subsidence (Fig. 5D–F).

Extreme summer SSTs in the Gulf are associated with lower atmospheric pressure over the Arabian Peninsula and higher pressure over Iran and the northern Arabian Sea (Fig.6A and Fig 6E). This pressure configuration weakens the Shamal winds and strengthens the monsoon winds over the southern and western Arabian Sea, resulting in reduced evaporation across most of the Gulf and enhanced evaporation over much of the Arabian Sea, respectively (Fig. 6C). Increased moisture transport from the Arabian Sea into the Gulf region, combined with reduced moisture export from the Gulf due to weakened local Shamal winds, leads to the accumulation of atmospheric moisture over the region (Fig. 6D). This, in turn, results in higher downward longwave radiation and reduced evaporative cooling, both of which contribute to elevated Gulf SSTs (Fig 3 and Fig S5, SI). In the upper troposphere (300 hPa), extreme Gulf SSTs are associated with increased convergence and enhanced subsidence over the region (Fig. 6G–H). The intensified upper-level subsidence contributes to heat trapping near the surface.

In summary, higher-than-average Gulf SSTs are associated with lower surface pressure over the Arabian Peninsula and higher pressure over Iran and Pakistan, leading to a weakening of the Shamal winds and a strengthening of the monsoon winds in the southern and western Arabian Sea (Fig 7). These conditions result in: (i) a buildup of atmospheric moisture over the Gulf, (ii) reduced evaporative cooling, and (iii) enhanced subsidence in the upper troposphere. Together, these three mechanisms contribute to heat being trapped near the surface and suppress surface cooling, thereby leading to warmer SSTs. Next, we examine how strongly these atmospheric conditions are tied to large-scale climate teleconnections.

**Figure 6.** Atmospheric conditions favoring summer MHWs in the Gulf. Composite of atmospheric conditions corresponding to SSTs 1.28 SD above the mean (90th percentile) over 75% of the Gulf. (A) Surface pressure, (B) air temperature at 2 m, (C) evaporation rate, (D) total column water vapor, (E) geopotential height and wind at 850 hPa, (F) vertical velocity at 850 hPa (negative values indicate upward motion), (G) geopotential height and wind at 300 hPa, and (H) vertical velocity at 300 hPa.

Figure 7. Illustration of atmospheric conditions causing cool and warm SSTs in the Gulf during summer. Schematic illustration of atmospheric conditions favoring cool (A) and warm (B) SSTs in the Gulf. Lower (higher) pressure over the Arabian Peninsula and higher (lower) pressure over Iran/Pakistan are associated with stronger (weaker) monsoon winds over the Arabian Sea, weaker (stronger) Shamal winds over the Gulf and stronger subsidence in the higher troposphere. These conditions lead to enhanced (reduced) evaporation over the Arabian Sea, increased (reduced) air moisture transport to the Gulf region, reduced (enhanced) evaporation over the Gulf and increased (reduced) retention of air moisture near the surface. Consequently, these changes enhance (reduce) heat trapping near the surface and weaken (increase) evaporative cooling at the surface.

Figure 8. Gulf SSTs and large-scale teleconnections. Zero-lag correlations between Gulf SSTs and major climate oscillation modes in summer (from left to right and top to bottom: ENSO, NAO, IOD, ISM). Hatching indicates statistical significance at 95% confidence interval.

# 3.3 Impact of Major Climate Variability Modes on Extreme Summer SSTs in the Gulf

We examine how summer Gulf SSTs are linked to four major modes of interannual variability that are likely to influence atmospheric circulation in the region: ENSO, NAO, IOD, and ISM.

We find that high summer SST anomalies in the Gulf are negatively correlated with ENSO, NAO, and IOD, and positively correlated with ISM (Fig. 8). However, only the correlations with ENSO and NAO are statistically significant at the 95% confidence level across most of the Gulf, whereas the correlations with IOD and ISM reach, at best, the 90% confidence level over the western and central Gulf, respectively. Moreover, while ENSO exhibits a negative correlation with SST throughout most of the summer (July–September) and leads the SST signal by 1 to 2 months, NAO is negatively correlated with SST only during August–September, with no discernible lead time (Fig. 9A and Fig. 9B). Similar to ENSO, IOD shows a persistent negative correlation with a 1-month lead (Fig. 9C). In contrast, ISM displays a positive correlation with SST, peaking at a 2-month lead during most of the summer (Fig. 9D). Interestingly, with the exception of late summer (September), the zero-lag correlation between ISM and SST is negative or very low, suggesting that higher SSTs are favored by strong monsoon winds in late spring to early summer (May–June), while stronger monsoon circulation in the peak summer season (July-August) has either little impact or even a suppressing effect on Gulf SSTs (Fig. 9D).

Figure 9. Lead-lag correlations between major climate variability modes and Gulf SST. Lead-lag correlations between ENSO (A), NAO (B), IOD (C) and ISM (D) and Gulf SST during summer months. Statistically significant correlations at 95% confidence interval are marked with white filled circles.

**Table 1.** Phases of major climate variability modes during years when more than 75% of the Gulf exhibits SST anomalies exceeding 1.28 standard deviations (90th percentile). Greyed years correspond to cases where SST anomalies exceed 2 standard deviations (98th percentile) over at least one-third of the Gulf. Symbols –, +, and N indicate negative, positive, and neutral phases, respectively. Positive and negative phases of ENSO correspond to El Niño and La Niña, respectively)

| Year | 1996 | 1998 | 1999 | 2000 | 2007 | 2010 | 2015 | 2016 | 2017 |
|------|------|------|------|------|------|------|------|------|------|
| ENSO | N    | -    | -    | -    | -    | -    | +    | _    | N    |
| NAO  | N    | -    | N    | -    | N    | -    | -    | -    | -    |
| IOD  | -    | -    | N    | N    | N    | N    | N    | -    | N    |
| ISM  | N    | N    | +    | +    | N    | N    | N    | N    | +    |

Six out of the nine summers in which SSTs exceeded the 90th percentile (i.e., 1.28 standard deviations above the mean) over more than three-quarters of the Gulf occurred during La Niña conditions (1998, 1999, 2000, 2007, 2010, and 2016; Table 1). During these extreme summers, the NAO was either in a negative phase (1998, 2000, 2010, 2015, 2016, 2017), or in a neutral phase (1996, 1999, 2007) (Table 1). The IOD, on the other hand, was mostly in a neutral phase (6 out of 9 summers) or in a negative phase (1996, 1998, 2016). Similarly, ISM was generally near its climatological mean (6 out of 9 summers) or above-average monsoon strength in the early summer (1999, 2000, 2017). When considering more extreme events—defined as SSTs exceeding the 98th percentile (i.e., 2 standard deviations above the mean) over more than one-third of the Gulf—2 out of the 3 such summers coincided with La Niña conditions (1998, 1999). Additionally, 1998, which recorded the highest interannual SST anomaly, also featured simultaneous negative phases of both NAO and IOD (Table 1). Other extreme SST summers that occurred in the absence of La Niña (2017, 1996, and 2015) were associated with strong negative IOD (1996), strong negative NAO (2015), or a combination of strong monsoon winds in early summer followed by weaker monsoon winds during peak summer along with a negative NAO in late summer (2017) (Table 1).

In summary, extreme summer SSTs in the Gulf are generally associated with La Niña conditions and a negative NAO phase, and to a lesser extent with negative IOD conditions and a monsoon pattern characterized by strong winds in early summer followed by weaker winds during peak summer. In the following section, we describe the mechanisms through which these teleconnections influence Gulf SSTs.

## 3.4 Mechanisms Linking Large-Scale Teleconnections to Gulf SST Variability

La Niña favors elevated summer SSTs primarily because it is associated with lower atmospheric pressure over the Arabian Peninsula at the surface and in the lower troposphere (Fig 10B). This pressure anomaly weakens the Shamal winds over the Gulf and strengthens the monsoon winds over the southern and western Arabian Sea (Fig 10D). The intensified winds over the Arabian Sea lead to enhanced evaporation (Fig 10F). Furthermore, the intensified low-pressure system over the Arabian Peninsula facilitates the transport of excess air moisture from the Arabian Sea to the Gulf region, trapping heat in the lower troposphere (Fig. 10H). Additionally, the weakened Shamal winds over the Gulf further amplify this warming by reducing evaporative cooling—and by enhancing the accumulation of atmospheric moisture in the region.

Negative NAO favors elevated summer SSTs in the Gulf primarily because it is associated with lower atmospheric pressure over the Arabian Peninsula and higher pressure over Iran and the northern Arabian Sea (Fig. 11B). This pressure pattern significantly weakens the surface pressure gradient across the Gulf, thereby substantially weakening the Shamal winds and reducing evaporation within the Gulf, while exerting limited influence on monsoon winds and evaporation over the Arabian Sea (Fig. 11D and Fig.11F). These changes contribute to higher Gulf SSTs by reducing evaporative cooling and enhancing the retention of atmospheric moisture over the region, which in turn traps heat near the surface (Fig. 11H).

Finally, the impacts of the ISM and IOD on summer Gulf SSTs are weaker and more complex. A strong monsoon circulation in early summer (i) enhances evaporation over the Arabian Sea, increasing atmospheric moisture in the region, and (ii) coincides with weaker Shamal winds over the Gulf during peak summer (July–August) (Fig. S6, SI). Both factors favor warmer Gulf SSTs during the peak summer season. In contrast, strong monsoon winds during peak summer (i) suppress evaporation over the

**Figure 10.** Composite anomalies during ENSO. Anomalies in summer atmospheric conditions during El Niño (left) and La Niña (right) over the Arabian Peninsula and Arabian Sea region. (A-B) sea level pressure, (C-D) wind speed, (E-F) evaporation rate, (G-H) specific humidity at 850 hpa.

**Figure 11.** Composite anomalies during NAO. Anomalies in summer atmospheric conditions during positive (left) and negative (right) NAO over the Arabian Peninsula and Arabian Sea region. (A-B) sea level pressure, (C-D) wind speed, (E-F) evaporation rate, (G-H) specific humidity at 850 hpa.

Arabian Sea due to upwelling-induced surface cooling, and (ii) tend to be associated with stronger Shamal winds over the Gulf (Fig. S6, SI). Together, these effects limit Gulf warming. Negative IOD events appear to be associated with slightly weaker Shamal winds and increased atmospheric moisture over the Gulf (Fig. S7, SI), both of which contribute to warmer SSTs. However, the magnitude of atmospheric anomalies associated with the IOD over the Gulf remains relatively weak compared to those linked to ENSO and NAO.

In summary, both La Niña and negative NAO phases favor elevated summer SSTs in the Gulf, primarily because they are associated with lower pressure over the Arabian Peninsula, which weakens the Shamal winds. However, La Niña is also linked to enhanced monsoon winds in the southern and western Arabian Sea, leading to greater moisture transport to—and accumulation in—the Gulf region. In contrast, a negative NAO is associated with a stronger weakening of the Shamal winds, resulting in a more pronounced reduction in evaporative cooling over the Gulf. A strong ISM in early summer followed by a weaker ISM during peak summer promotes increased moisture buildup in the Gulf, while a negative IOD contributes slightly to warmer Gulf SSTs due to marginally weaker Shamal winds.

## 3.5 Cumulative Impacts of Climate Variability Modes on Gulf Summer SSTs

The climate variability modes considered in this study are not independent but can interact, resulting in cumulative or nonlinear effects. In particular, ENSO, IOD, and the ISM exhibit strong interdependence (Kirtman and Shukla, 2000; Behera et al., 2006; Ashok and Saji, 2007; Cai et al., 2011). Although their relationship is more complex, ENSO and NAO can also interact and produce synergistic effects (Wallace and Gutzler, 1981; Jiménez-Esteve and Domeisen, 2018; Xu et al., 2024), albeit much more weakly during summer (Zhang et al., 2019). Our cross-correlation analysis confirms these links, showing that ENSO is significantly correlated with both the IOD  $(r=0.36,\ p<0.01)$  and the ISM  $(r=0.38,\ p<0.01)$ , while its correlation with the summer NAO is not statistically significant  $(r=0.09,\ p>0.05)$  (Table 2). This weak summer ENSO-NAO coupling is consistent with Folland et al. (2009), who reported an asymmetric summer relationship, with La Niña episodes weakly associated with negative NAO phases, but no significant link between El Niño and positive NAO phases. To isolate the unique relationship between Gulf summer SST and each of the four climate modes while controlling for the influence of the others, we conducted a partial regression analysis (Figure S8, SI). This analysis shows that both NAO and ENSO are significantly and negatively correlated with summer Gulf SSTs, whereas IOD and ISM display only weak and statistically insignificant correlations.

To further quantify the collective influence of these modes, we applied a stepwise multiple linear regression analysis (Figure 12). The results indicate that ENSO and NAO together explain more than 50% of the total SST variability, while the addition of IOD and ISM increases the explained variance only marginally. This limited contribution reflects both their weaker individual correlations with SST and their strong association with ENSO, which reduces their independent explanatory power (Table 2). A decomposition of the total explained variance using hierarchical partitioning (Mac Nally, 1996) further confirms the dominance of NAO and ENSO, with the NAO showing a somewhat stronger unique contribution owing to its weaker correlations with the other three modes (Fig S9, SI). The analysis also reveals that approximately 20% of the total variance is jointly explained by all four modes, highlighting the intertwined nature of these large-scale climate drivers.

**Figure 12. Explained variance by regression models of Gulf SST as a function of predictors.** Summer (July-September) SST variance explained by different combinations of climate variability modes. Note that ENSO and NAO together account for more than 50% of the total SST variability. In contrast, the inclusion of IOD and ISM contributes only marginally to the explained variance.

**Table 2.** Correlation matrix of variability modes used in the regression model. Correlations with the asterisk sign (\*) are statistically significant at 95% confidence interval.

|      | ENSO | NAO   | IOD    | ISM    |
|------|------|-------|--------|--------|
| ENSO | 1    | 0.093 | 0.361* | 0.381* |
| NAO  | _    | 1     | -0.005 | -0.043 |
| IOD  | _    | _     | 1      | -0.018 |
| ISM  | _    | _     | _      | 1      |

To better visualize the impact of the interaction of the climate modes on SST, and how it manifested in specific years, we use the Self-Organizing Maps (SOM). Since ENSO and NAO together explain most of the variance in Gulf SST, we applied the SOM analysis to three variables: Gulf SST as the dependent variable, and ENSO and NAO as the independent variables or predictors (Fig 13).

The distribution of SST anomalies and ENSO/NAO phases on the SOM reveals four Gulf SST regimes, corresponding to four quadrants on the SOM grid: (i) very warm summers where La Niña coincides with a negative NAO such as in 1998, 2000, 2010 (upper-right quadrant); (ii) very cool summers where El Niño coincides with a positive NAO such as in 1982, 1991, 2018 (lower-left quadrant); (iii) average to moderately warm summers where La Niña coincides with a positive NAO such as in 1981, 2020, 2022 (upper-left quadrant) and (iv) average to moderately warm summers where El Niño coincides with a negative NAO such as in 1986, 1987, 2015 (lower-right quadrant) (Fig 13).

Figure 13. Visualization of Gulf SST links to ENSO and NAO using a Self-Organizing Map (SOM). Component planes of the SOM. Each plane shows the distribution of one variable across the map. Each neuron corresponds to a prototype of Gulf SST regime defined by a given combination of an SST anomaly and the corresponding phases of ENSO and NAO. Note that ENSO and NAO act mostly as two independent variables along which a large fraction of the SST can be explained. The map is divided into 4 quadrants that correspond to 4 regimes: (i) an upper right quadrant with the warmest summers where La Niña coincides with negative NAO, (ii) a lower-left quadrant with the coolest summers where El Niño coincides with positive NAO, and (iii) the upper-left quadrant and lower-right quadrant where La Niña coincides with positive NAO and El Niño coincides with negative NAO, respectively. Individual years are shown associated with their best-matching neurons.

Figure 14. Contrasting impacts of climate teleconnections on SSTs in the Gulf and Arabian Sea. Composite SST anomalies showing the difference between opposite phases of major climate oscillation modes during summer (from left to right and top to bottom: ENSO, NAO, IOD, ISM).

In summary, since ENSO and NAO are largely uncorrelated (i.e., orthogonal), their effects on Gulf SSTs appear to be predominantly additive, with each mode independently contributing to the overall SST anomaly.

## 3.6 Divergent Thermal Responses of the Gulf and Arabian Sea to Major Climate Modes

ENSO, NAO, IOD, and ISM exert distinct influences on surface temperatures in the Gulf and the Arabian Sea (including the Sea of Oman) (Fig. 14). For example, while La Niña, negative NAO, negative IOD, and strong early-summer monsoon circulation are all associated with extreme summer temperatures in the Gulf, these same climate patterns tend to either favor cooling over much of the Arabian Sea—including the northern Arabian Sea and the Sea of Oman (in the case of the IOD and ISM)—or produce only limited warming, confined to specific areas, such as the northern Arabian Sea (La Niña) or the eastern Arabian Sea (negative NAO), with cooling over the remaining regions.

These contrasting responses between the Gulf and the Arabian Sea are noteworthy, as several previous studies (e.g., Al-Rashidi et al., 2009; Nandkeolyar et al., 2013; Bordbar et al., 2024) have assumed that Gulf SSTs respond similarly to those of the adjacent Arabian Sea under large-scale teleconnections. In contrast, our results reveal an anti-correlated behavior between

the two basins. For instance, while positive IOD and El Niño events are typically linked to warming across the Arabian Sea, they coincide with weak to moderate cooling in the Gulf.

These contrasts likely arise from two main factors: (i) differences in how these modes modulate local winds—positive IOD and El Niño events are associated with weaker winds over the Arabian Sea but a modest strengthening of the Shamal winds over the Gulf (Fig. 10; Fig. S7, SI); and (ii) stronger role of ocean circulation including upwelling and lateral heat advection in the Arabian Sea relative to the Gulf, due to the shallow and semi-enclosed nature of the latter, which limits the propagation of temperature anomalies from the Arabian Sea into the Gulf, especially in summer (Lachkar et al., 2024).

These opposing SST responses enhance the temperature gradient between the Gulf and the Arabian Sea during summer, potentially weakening the density contrast and, consequently, the water exchange between the two seas. Such a reduction in exchange could have important biogeochemical implications for both systems, affecting nutrient fluxes, oxygen concentrations, and ecosystem productivity (Lachkar et al., 2021, 2024).

## 4 Discussion

## 4.1 Gulf SST Extremes in the Context of Global MHW Drivers

Our study reveals that extreme summer SSTs in the Gulf are typically associated with both local climatic changes (e.g., weakening of the Shamal winds) and remote large-scale influences (e.g., modulation by climate modes such as ENSO and the intensity of the Indian monsoon circulation).

These findings align with previous research on MHWs, indicating that extreme SSTs often arise from a combination of local-scale processes and large-scale teleconnections (Holbrook et al., 2019). Regarding local-scale drivers, our results align with those from studies of other semi-enclosed seas, which highlight the role of anomalous atmospheric conditions in the development of MHWs (Gröger et al., 2024), particularly during warmer months and MHW onset periods (Denaxa et al., 2024). Specifically, the weakening of winds and the resulting reduction in surface heat loss via evaporative cooling have been shown to contribute to MHWs in various oceanic settings (e.g., Yao et al., 2020; Benthuysen et al., 2018; Gröger et al., 2024). However, many of these studies also link MHWs to increased shortwave solar radiation due to reduced cloud cover. In contrast, our findings suggest that Gulf SSTs increase in summer despite reduced shortwave radiation. This is due to enhanced atmospheric moisture, which increases haze and reduces incoming solar radiation, while simultaneously enhancing longwave radiation and trapping heat near the surface. On the large-scale teleconnection side, studies in tropical and subtropical regions have shown that ENSO and IOD strongly influence the occurrence of MHWs in the tropical Indian Ocean, while ENSO and the NAO exert strong influence over MHWs in the Pacific and Atlantic Oceans, respectively (Oliver et al., 2018; Holbrook et al., 2019). However, in contrast to these findings where El Niño events are typically associated with increased MHW activity in the tropical Indian and Pacific Oceans, our study shows that La Niña conditions are more strongly associated with higher Gulf summer SSTs. Similarly, while increased MHW occurrences in the western Indian Ocean are often linked to positive IOD phases, we find that warm summer SSTs in the Gulf are generally associated with neutral or negative IOD phases.

## 4.2 Caveats and Limitations

To assess the sensitivity of our results to the choice of reanalysis product, we repeated the analyses using two additional atmospheric datasets: JRA-55 and MERRA-2. Consistent with the ERA5-based results, composites of atmospheric conditions corresponding to extreme summer SSTs (90th percentile) across most of the Gulf, derived from these alternative products, display similar large-scale patterns (Fig S10, SI). Specifically, both JRA-55 and MERRA-2 show that elevated Gulf SSTs are associated with lower atmospheric pressure over the Arabian Peninsula and higher pressure over Iran and the northern Arabian Sea, accompanied by a weakening of the Shamal winds, a strengthening of the monsoon flow over the southern and western Arabian Sea, and enhanced surface air temperature and total-column water vapor over the Gulf and the Arabian Peninsula (Fig. S10, SI). Similarly, analyses of summer atmospheric conditions associated with major climate variability modes (ENSO and NAO) were repeated using JRA-55 and MERRA-2 (Fig. S11 and Fig S12, SI). The resulting patterns are consistent across all three reanalysis products. As with ERA5, La Niña (El Niño) conditions are associated with lower (higher) atmospheric pressure over the Arabian Peninsula (Fig S11, SI), while negative (positive) NAO phases correspond to lower (higher) pressure over the Arabian Peninsula and higher (lower) pressure over Iran and the northern Arabian Sea (Fig S12, SI).

Overall, these comparisons indicate that the large-scale atmospheric circulation patterns associated with extreme summer SSTs in the Gulf are robust across multiple reanalysis datasets. Nonetheless, all three reanalyses share a relatively coarse spatial resolution, which may limit their ability to capture small-scale features such as the effects of complex orography. These limitations are further compounded by the scarcity of long, continuous meteorological observations in the region, which constrains the validation of reanalysis-based fields at local scales.

While this study establishes robust statistical links between Gulf SST anomalies and major climate variability modes such as ENSO and the NAO, primarily through their associated modulation of surface pressure and wind patterns over the Arabian Peninsula and Iran, it does not explicitly examine the underlying physical mechanisms by which these remote modes influence regional atmospheric circulation. Understanding the dynamical pathways that connect Pacific and North Atlantic variability to the Arabian Peninsula's atmospheric conditions remains a complex problem that extends beyond the scope of the present analysis, which focuses on characterizing SST variability within the Gulf itself. Several previous studies have reported that ENSO and NAO modulate pressure systems over the Arabian Peninsula and Iran without fully explaining the mechanisms responsible for this modulation (e.g., Folland et al., 2009; Chronis et al., 2011; Yu et al., 2016). Other studies (e.g., Niranjan Kumar and Ouarda, 2014; Attada et al., 2019; Cheng et al., 2023) have proposed potential teleconnection pathways involving large-scale Rossby wave trains or adjustments of the subtropical jet stream in the upper troposphere. Nevertheless, a comprehensive dynamical attribution has yet to be established. Future work combining observational analyses with targeted climate-model experiments will be required to elucidate these linkages in greater detail.

Finally, this study focuses on understanding extreme SSTs at the scale of the entire Gulf. Our large-scale analysis reveals that most of the interannual variability is driven by fluctuations in local atmospheric conditions, with only a limited contribution from remote oceanic influences. However, at more localized scales—particularly near the Strait of Hormuz—the influence of

oceanic circulation and heat transport from the Sea of Oman, and thus changes within the Sea of Oman itself, may become more significant and warrant further detailed investigation.

# 4.3 Implications for the Predictability of Coral Bleaching in the Gulf

We show that summer Gulf SSTs are strongly influenced by large-scale climate modes such as the NAO and ENSO. Consequently, the predictability of these teleconnections can enhance the predictability of summer SSTs in the Gulf.

While summer NAO predictability remains limited—owing to the weaker North Atlantic jet stream and reduced ocean–atmosphere coupling during summer—previous studies have demonstrated moderate skill using coupled models, particularly when initialized in late spring (Folland et al., 2009; Gastineau and Frankignoul, 2015; Dunstone et al., 2023).

In contrast, ENSO generally exhibits higher and improving predictability. However, forecasts are challenged by the spring predictability barrier (March–May), a period marked by weakened ocean–atmosphere coupling (McPhaden, 2003). Forecast skill improves in late spring as the barrier subsides and model performance in tracking ENSO evolution through summer increases (Duan and Wei, 2013; Chen et al., 2020), particularly for strong events associated with pronounced subsurface heat content anomalies in the equatorial Pacific (Zhang et al., 2024).

Therefore, subseasonal-to-seasonal forecasts that incorporate North Atlantic and equatorial Pacific precursors may provide skill in predicting summer Gulf SSTs with 2-3 month lead time when initialized in late spring (e.g., May). This predictive skill is likely enhanced during strong ENSO and NAO phases, offering potential for early warning of summer MHWs in the region.

As coral reefs represent the most biodiverse ecosystem in this arid region, with significant economic importance in supporting fisheries and ecotourism (Vaughan et al., 2019), the ability to predict anomalously hot summers weeks to months in advance would be of great value. Mass coral bleaching associated with recurrent marine heat waves over the past several decades have resulted in a loss of 40% of coral across the Gulf since the mid-1990s, largely because there was no advanced warning to permit preventative measures (Burt et al., 2020). Having forecasts of imminent heat risks would allow regional reef managers and other practitioners to proactively deploy approaches to minimize coral bleaching. These include rapid interventions such as reef shading (Coelho et al., 2017), pumping of deep, cool water onto reefs (Baird et al., 2020), or artificial mixing to decrease stratification (Harrison, 2024), as well as using management efforts to reduce additive stressors that exacerbate heat stress (e.g. by limiting dredging activities or reducing nutrient inputs during peak heat periods, Gove et al., 2023). Thus, early warning forecasts provide the capacity for rapid adaptive management intervention in ways that were not previously available.

## 5 Conclusions

While climate change is driving a rapid long-term increase in Gulf SSTs (e.g., Lachkar et al., 2024), natural climate variability modulates this warming trend on multiple timescales. In this study, we analyzed the interannual variability of Gulf SSTs over recent decades, with a focus on identifying the dominant physical drivers.

Our analysis reveals that elevated Gulf summer SSTs are primarily associated with three key processes: (i) accumulation of atmospheric moisture, driven by weakened Shamal winds and enhanced moisture transport from the Arabian Sea, which traps

heat in the lower troposphere; (ii) reduction of evaporative cooling, also linked to the suppression of the Shamal winds; (iii) enhanced upper-tropospheric subsidence, which further contributes to trapping heat near the surface (Fig 7). These conditions occur most frequently during La Niña conditions or in conjunction with negative NAO phases. The most pronounced warm SST anomalies are observed when La Niña and negative NAO phases co-occur (e.g., 1998). Collectively, ENSO and NAO account for more than 50% of the interannual variability in Gulf SSTs.

These findings carry important implications for the seasonal predictability of summer MHWs in the Gulf. Since large-scale climate modes exert a strong influence on Gulf SSTs—and typically precede SST anomalies—there is promising potential for seasonal forecasting of MHW risk. Such forecasts could underpin early warning systems and support mitigation efforts, particularly to protect vulnerable marine ecosystems such as coral reefs. This potential will be explored further in future research.

Code and data availability. The ROMS model code is available in Auclair et al. (2018). The model outputs are described and available in Lachkar (2024). The atmospheric forcing data is available in European Centre for Medium-Range Weather Forecasts (2012). The lateral boundary conditions is available in Copernicus Climate Change Service, Climate Data Store (2021). ERA5 reanalysis data was retrieved from the Copernicus Climate Change Service (C3S) Climate Data Store (CDS). The ERA5 data is documented in Hersbach et al. (2023).

Author contributions. ZL conceived and designed the study, carried out the data analysis and wrote the manuscript with contributions from OP, FP, BK and JB. All authors contributed to the article and approved the submitted version.

*Competing interests.* The authors declare no competing interests...

Acknowledgements. This research was supported by funding provided by Tamkeen through grants CG009 to the Mubadala ACCESS center and CG007 to the Water Research Center, as well as funding support from Mubadala Philanthropies under XR016; their support is greatly appreciated. Computations were performed at the High Performance Computing (HPC) cluster of NYUAD, Jubail. We extend thanks to the NYUAD HPC team for technical support.

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
