# Peer review of "Local and remote climatic drivers of extreme summer sea surface temperatures in the Arabian Gulf"

_EGUsphere, 2025_

## Author Comment (AC1)

**Response to Reviewer #1**

Please note that the reviewer's comments are shown in *blue*, while our responses are shown in *red*.

**RC1: 'Comment on egusphere-2025-2948', Anonymous Referee #1, 17 Jul 2025 Comments:**

This study examines sea surface temperature (SST) variability in the Persian Gulf and its relationship with large-scale climate patterns (ENSO, NAO and IOD). The authors indicated that local atmospheric anomalies significantly impact SSTs in the Gulf by modulating heat fluxes. ENSO, NAO and IOD could impact SSTs in the Gulf by modulating the local atmosphere circulation. The combined effect of ENSO and NAO on SSTs in the Gulf is also discussed. The results obtained in this study are interesting. This manuscript can be accepted after revisions.

We sincerely thank the reviewer for their positive and encouraging feedback, and for the time and effort they invested in reviewing our manuscript.

1) To confirm the results obtained from the ERA5, the authors should use other reanalysis data (e.g. MERRA2 and JRA55).

Following the reviewer's recommendation, we will present the local atmospheric conditions associated with extreme summer SSTs in the Gulf, as well as their links with large-scale climate variability modes such as ENSO and the NAO, using the two additional reanalysis datasets (MERRA-2 and JRA-55) suggested by the reviewer.

In the revised manuscript, we will show that our key results based on ERA5, namely that extreme summer Gulf SSTs are favored by lower-than-normal air pressure over the Arabian Peninsula and higher-than-normal pressure over Iran and Pakistan, leading to a weakening of the Shamal winds, are consistently reproduced in both additional reanalyses. The associated higher atmospheric moisture over the Gulf region, as well as the role of La Niña and negative NAO phases in promoting these conditions, are also captured in MERRA-2 and JRA-55. This consistency across three independent datasets strengthens confidence in the robustness of our findings with respect to the choice of reanalysis product.

Given the higher spatial resolution of ERA5 (¼°) compared to the other datasets, whose resolutions are two to five times coarser, and considering the critical role of horizontal resolution in representing regional processes over the Arabian Peninsula and the Gulf, we will keep the detailed analysis focused on ERA5 and the regional hindcast simulation. We will, however, show selected diagnostics from all three reanalysis products in the Supplementary Information to demonstrate the robustness of our main conclusions.

2) It should be noted that ENSO, IOD and NAO are not independent of each other. For instance, ENSO can affect IOD, IOD can affect ENSO, and NAO can also affect ENSO. The authors should examine the combined and relative effects of ENSO, IOD and NAO on SSTs in the Gulf using partial composite or regression analysis.

We agree with the reviewer that these climate modes are not independent but rather interact with one another. In the revised manuscript, we will add citations to some key references on the connection between these variability modes. In the correlation matrix presented in the original manuscript (Table 2, page 20), we showed that ENSO is significantly correlated with both the IOD (r = 0.36, p < 0.01) and the ISM (r = 0.38, p < 0.01), whereas its correlation with the summer NAO is not statistically significant (r = 0.09, p > 0.05).

This realization motivated our decision to perform a multiple linear regression analysis that accounts for the influence of each predictor while controlling for the others (Fig. 11 in the original manuscript). To further clarify this point and in line with the reviewer's suggestion, we will additionally conduct a partial regression analysis to show the correlation coefficients and significance levels between Gulf SST and each predictor while controlling for the remaining predictors.

Furthermore, we will apply a hierarchical partitioning approach (Mac Nally, 1996, *Australian Journal of Ecology*, 21, 224–228) to quantify the relative importance of each predictor in the multiple regression framework. This method is particularly suitable when predictors are correlated, as it is independent of the order in which variables enter the model. It decomposes the total explained variance into the unique contribution of each predictor and the variance jointly explained with other correlated predictors (shared contributions). This additional analysis reveals that, owing to its weaker correlations with the other three modes, the NAO exhibits a higher unique contribution to the explained variance compared to ENSO, which shows stronger associations with the IOD and ISM. A substantial fraction (about 20%) of the total explained variance is shared among all four modes.

3) In addition to ENSO and the NAO, I would suggest that the authors also examine the role of Arctic sea ice anomalies in shaping extreme temperatures in the Gulf. Recent studies have indicated that Arctic sea ice anomalies could have a significant impact on ENSO, NAO and IOD. I suggest add some discussions. (https://doi.org/10.1175/JCLI-D-24-0419.1.; https://doi.org/10.1038/s41612-025-00936-x.; https://doi.org/10.1175/JCLI-D-23-0733.1.).

In response to the reviewer's suggestion, we examined potential links between summer Gulf SST and both Arctic sea ice and the Arctic Oscillation (Figs. 1 and 2). We found no statistically significant correlations between Gulf SST and either variable.

Fig 1. Zero-lag (left) and 6-month-lag (right) correlation between summer Gulf SSTs and Arctic sea ice area anomaly. Correlations are not statistically significant.

Fig 2. Zero-lag (left) and 6-month-lag (right) correlation between summer Gulf SSTs and the Arctic Oscillation index. Hatching indicates statistical significance at 95% confidence interval.

We conclude that, despite the potential connections between Arctic sea ice and the development of IOD and ENSO highlighted in the studies cited by the reviewer, these links do not appear to play a critical role in determining Gulf summer SSTs. This may be because these relationships are weaker during summer or because their influence does not substantially impact the summer atmospheric circulation over the Gulf region. Moreover, given the strong coupling between NAO and AO, we consider that any potential influence of the AO on Gulf summer SST is likely already captured by the NAO signal. These points will be briefly discussed in the revised manuscript.

4) In addition to atmospheric heat fluxes, the tendency of SST should also be significantly impacted by oceanic dynamics, such as advection and upwelling/downwelling. The authors should analyse the SST tendency equation and investigate the role of these oceanic processes.

We agree with the referee that the tendency of SST results from the balance between atmospheric heat fluxes and oceanic circulation and transport (mixing and advection). However, we argue that, due to the shallow (

Fig 3. Interannual anomalies of the temperature tendency term (black), atmospheric heat fluxes (red), and vertical (blue) and horizontal (green) heat transport fluxes, integrated over the entire Gulf. Because the tendency term and horizontal transport fluxes are much smaller than the atmospheric heat fluxes and vertical transport fluxes, the former two are plotted in °C per year, while the latter two are shown in °C per month, allowing all four terms to be displayed on the same scale.

5) The authors discussed the possible influence of ENSO, NAO and IOD on SST anomalies in the Gulf. However, the underlying physical mechanisms have not been investigated in detail. The authors should examine the physical processes through which ENSO and NAO impact the formation of local atmospheric circulation. For example, Cheng et al. (2023;

https://doi.org/10.1007/s00382-022-06616-3) indicated that AO/NAO-related atmospheric heating over the North Atlantic could trigger an atmospheric wave train from the North Atlantic to the northern Indian Ocean.

Understanding the mechanisms of global teleconnections that link large-scale climate variability modes to the modulation of regional summer atmospheric circulation over the Arabian Peninsula and Gulf region is an important open research question. However, a detailed investigation of these mechanisms is beyond the scope of the current study. This limitation will be highlighted in a new section, "Caveats and Limitations," in the revised manuscript.

Nevertheless, the revised manuscript will include a brief discussion of mechanisms proposed in studies that have explored aspects of this question. For example, Cheng et al. (2023) proposed that atmospheric heating anomalies over the North Atlantic can trigger a Rossby wave train toward the northern Indian Ocean, potentially affecting the development of the Indian Ocean Dipole (IOD) in autumn. However, as our results show that the IOD has a limited impact on Gulf summer SST, likely due to the minor influence of Indian Ocean heat advection on interannual Gulf SST variability, this mechanism may contribute only marginally to summer Gulf SST variability.

Several studies have examined the connection between ENSO and summer atmospheric circulation in the region. For instance, Yu et al. (2015) identified a statistically significant relationship between ENSO and the timing of onset and termination of summer Shamal winds, which they linked to the Iranian summer heat low. However, that study did not propose mechanisms explaining the linkage between tropical Pacific SST anomalies and the development of the Iranian heat low. Attada et al. (2019) linked a strong Indian Summer Monsoon, generally favored during La Niña conditions, to increased adiabatic warming over the region due to enhanced subsidence in the middle and upper troposphere. This is consistent with our findings, which show increased subsidence in the upper troposphere over the Arabian Peninsula.

Finally, Kuman and Ouarda (2014) investigated the link between UAE winter precipitation and ENSO. They suggested that ENSO-driven changes in regional atmospheric circulation over the Arabian Peninsula are associated with shifts in planetary Rossby waves generated in the central Pacific due to SST anomalies, which then propagate into extratropical latitudes as cyclonic and anticyclonic features in the upper troposphere near the subtropical jet stream. While that study focused on winter conditions, similar mechanisms may contribute to ENSO-related changes in regional atmospheric circulation over the Arabian Peninsula during summer.

Regarding mechanisms linking NAO to regional atmospheric circulation, fewer studies have examined this relationship, particularly in summer. Chronis et al. (2011) showed that negative

summer NAO is associated with lower cloudiness and higher temperatures over the eastern Mediterranean and the Anatolian Plateau. Folland et al. (2009) demonstrated that negative summer NAO is associated with lower pressure over the northern Arabian Peninsula and Iraq. Such weakening of the high-pressure system that typically dominates the eastern Mediterranean and western Arabian Peninsula in summer can reduce the pressure gradient across the Gulf and, consequently, decrease the frequency and intensity of Shamal winds.

---

## Author Comment (AC2)

**Response to Reviewer #2**

Please note that the reviewer's comments are shown in *blue*, while our responses are shown in *red*.

**RC2: 'Comment on egusphere-2025-2948', Anonymous Referee #2, 03 Sep 2025**

The manuscript by Lachkar et al. addresses an important and timely topic by discussing the drivers of extreme summer sea surface temperatures in the Arabian Gulf. The authors combine an eddy-resolving ocean hindcast with ERA5 reanalysis, providing a framework to separate local and remote atmospheric influences. The work is well-motivated, as the Gulf is a critical hotspot of marine heat stress, with major ecological and socio-economic consequences. The manuscript is generally well-structured, the figures are clear, and the results are presented in a way that advances understanding of both local mechanisms (heat fluxes, Shamal winds) and large-scale teleconnections (ENSO, NAO). The finding that ENSO and NAO together explain over 50% of Gulf SST variability is particularly noteworthy and of interest for predictability. Overall, the paper makes a valuable contribution and is suitable for publication after some clarifications and refinements.

We sincerely thank the reviewer for their positive and encouraging feedback, and for the time and effort they invested in reviewing our manuscript.

**Specific Comments**

1) The current manuscript however distinguishes itself from earlier studies by focusing on summer SST extremes and their mechanistic links to atmospheric circulation, but the introduction could better highlight how this work fills the gap left by earlier Gulf studies that relied on multi-seasonal or coarser analyses. Also, how the Arabian gulf understanding can be important for precipitation extreme changes having said that arabian gulf as one of the moisture source for heavy rainfall (can be seen in Pathak et al. 2025).

Pathak, R., Dasari, H.P., Ashok, K. *et al.* Dynamics of intensification of extreme precipitation events over the Arabian Peninsula derived from CMIP6 simulations. *npj Clim Atmos Sci* 8, 126 (2025). https://doi.org/10.1038/s41612-025-01016-w

Following the referee's recommendation, we will include a reference highlighting the relevance of Gulf summer SST to extreme precipitation events in the region, given the role of the Gulf as a key

source of atmospheric moisture. We will also more clearly emphasize how the present study addresses gaps in earlier Gulf studies that relied on multi-seasonal data and coarser analyses.

2) It is interesting to see that ENSO and NAO act largely independently and additively is interesting. Would be interesting to have a discussion expanded to compare with other ocean basins, where ENSO–NAO interactions are sometimes nonlinear.

While the interactions among different climate variability modes are complex and influenced by many factors, thus extending beyond the scope of the current study, a brief discussion of this aspect will be added to the revised manuscript. In particular, we will cite Folland et al. (2009), who report a weak correlation between La Niña and negative summer NAO, but no significant correlation between positive NAO and El Niño.

3) At LN170 and other places. The analysis finds a weak role for IOD and ISM. Given that the IOD is often invoked in regional SST variability, it would be helpful to provide a more explicit explanation of why its influence is muted in the Gulf, possibly due to the strong dominance of atmospheric fluxes over advective processes.

We thank the reviewer for this observation. As noted by the referee (and discussed in our response to Reviewer 1's comment #4), interannual variability in summer Gulf SSTs is primarily driven by variability in atmospheric fluxes, with only a modest contribution from lateral ocean advection from the Arabian Sea. Although Arabian Sea SST variability is influenced by the IOD and ISM, the propagation of this signal from the western Arabian Sea through the Strait of Hormuz into the Gulf has a limited impact on Gulf summer SSTs. This explains why the effects of all major climate modes, including ENSO and NAO, on SSTs in the Gulf and the Arabian Sea are either weakly correlated or even strongly anticorrelated (see Figure 13). This point will be more clearly highlighted in the revised manuscript.

4) Please provide a schematic summarizing the coupled processes (weakening Shamal winds, enhanced subsidence, moisture build-up), and the same would be very useful for readers.

Such a schematic (Fig. 14) was already included in the revised manuscript, but it was previously only referenced in the Conclusion, which we recognize may not be the most appropriate location.

Therefore, in the revised manuscript, we will refer to the figure earlier, at the end of the Results section, to increase its visibility.

5) Since as we know that ENSO and NAO are routinely monitored, please provide lead time with which Gulf summer SST anomalies could be anticipated.

As discussed in Section 4.4, there is potential for seasonal forecasting of summer Gulf SSTs with a 2–3 month lead time when initialized in late spring (e.g., May), particularly once the spring predictability barrier of ENSO (March–May) subsides and the skill of summer NAO prediction improves (Dunstone et al., 2023). We will clarify this point further in the revised manuscript.

---

## Author Response (AR1)

We sincerely thank all reviewers for the time and effort they dedicated to evaluating our manuscript. Reviewers' comments are shown in blue, our responses in black, and modifications made to the manuscript text are shown in red.

In the revised manuscript, based on the reviewers' feedback and suggestions, we have:

- 1. **Added new diagnostics** presented in six new figures (Fig. S1 and Figs. S8–S12, Supplementary Information) to explore a potential teleconnection with the Arctic Sea (Fig. S1), quantify the unique contributions of each teleconnection to SST variability (Figs. S8 and S9), and assess the sensitivity of our results to the choice of atmospheric reanalysis product (Figs. S10–S12).
- 2. Improved the presentation of the **link between heat fluxes and SST anomalies** by including the surface heat budget based on the SST tendency equation, described in the new Section 2.2 and presented in Figure 2A.
- 3. **Added a dedicated section on limitations**, Section 4.2 (*Caveats and Limitations*), to discuss the main limitations of the study and aspects that warrant further investigation in future work.
- 4. Reorganized the manuscript for clarity and readability, including:
  - Moving the important schematic previously shown as Figure 14 (now Figure 7) earlier in the manuscript.
  - Relocating the previous Sections 4.1 and 4.2, where new results are presented, from the Discussion to the Results section (now Sections 3.5 and 3.6).
  - Improving clarity and consistency throughout the manuscript, including the Introduction.

Below, we **respond point-by-point** to each reviewer comment and detail how we revised the manuscript to address them. We believe the manuscript has been substantially improved as a result of these modifications, and we thank the three reviewers for their valuable feedback, which greatly strengthened the work.

**RC1: 'Comment on egusphere-2025-2948', Anonymous Referee #1, 17 Jul 2025 Comments:**

This study examines sea surface temperature (SST) variability in the Persian Gulf and its relationship with large-scale climate patterns (ENSO, NAO and IOD).. The authors indicated that local atmospheric anomalies significantly impact SSTs in the Gulf by modulating heat fluxes. ENSO, NAO and IOD could impact SSTs in the Gulf by modulating the local atmosphere circulation. The combined effect of ENSO and NAO on SSTs in the Gulf is also discussed. The results obtained in this study are interesting. This manuscript can be accepted after revisions.

We sincerely thank the reviewer for their positive and encouraging feedback, and for the time and effort they invested in reviewing our manuscript.

1) To confirm the results obtained from the ERA5, the authors should use other reanalysis data (e.g. MERRA2 and JRA55).

Following the reviewer's recommendation, we assessed the sensitivity of our results to the choice of reanalysis product by repeating key analyses using two additional atmospheric datasets: the Japanese 55-year Reanalysis (JRA-55; Kobayashi et al., 2015) and the Modern-Era Retrospective Analysis for Research and Applications, Version 2 (MERRA-2; Gelaro et al., 2017). Specifically, we evaluated composites of atmospheric conditions associated with extreme summer SSTs (90th percentile) across the Gulf and analyzed summer atmospheric conditions linked to major climate variability modes (ENSO and NAO), derived from these alternative products. The results obtained from JRA-55 and MERRA-2 were then compared with those based on ERA5.

These analyses are presented in three new figures (Figs. S10–S12) in the Supplementary Information (SI). Consistent with ERA5, both JRA-55 and MERRA-2 indicate that elevated Gulf SSTs are associated with lower atmospheric pressure over the Arabian Peninsula and higher pressure over Iran and the northern Arabian Sea, accompanied by a weakening of the Shamal winds, a strengthening of the monsoon flow over the southern and western Arabian Sea, and enhanced surface air temperature and total-column water vapor over the Gulf and the Arabian Peninsula (Fig. S10, SI). Similarly, analyses of summer atmospheric conditions associated with ENSO and NAO (Figs. S11–S12, SI) yield consistent patterns across all three reanalysis products: La Niña (El Niño) conditions correspond to lower (higher) atmospheric pressure over the Arabian Peninsula (Fig. S11,

SI), while negative (positive) NAO phases are linked to lower (higher) pressure over the Arabian Peninsula and higher (lower) pressure over Iran and the northern Arabian Sea (Fig. S12, SI).

While we retain the detailed analysis based on ERA5 (given its higher spatial resolution of ¼° compared to the other datasets, whose resolutions are two to five times coarser) and the regional hindcast simulation, these comparisons demonstrate that the large-scale atmospheric circulation patterns associated with extreme summer SSTs in the Gulf are robust across multiple reanalysis datasets, thereby reinforcing confidence in the robustness of our findings with respect to the choice of reanalysis product.

The new diagnostics and corresponding results have been introduced in the revised manuscript as follows:

- 1) We introduce the two products in the Method section, section **2.4, lines 134-138** of the revised manuscript: "Finally, to assess the robustness of the key findings with respect to the choice of reanalysis product, we additionally use two independent atmospheric reanalysis datasets: the Japanese 55-year Reanalysis (JRA-55; Kobayashi et al., 2015) and the Modern-Era Retrospective Analysis for Research and Applications, Version 2 (MERRA-2; Gelaro et al., 2017). Data from these datasets were extracted over the same study period and include the following variables: sea level pressure, 10-m winds, 850 hPa winds and geopotential height, and total-column water vapor."
- 2) We present and discuss the results of the comparison between the three reanalysis products in the Discussion section, in the new section **4.2 titled: "Caveats and Limitations", lines 410-426** of the revised manuscript:

"To assess the sensitivity of our results to the choice of reanalysis product, we repeated the analyses using two additional atmospheric datasets: JRA-55 and MERRA-2. Consistent with the ERA5-based results, composites of atmospheric conditions corresponding to extreme summer SSTs (90th percentile) across most of the Gulf, derived from these alternative products, display similar large-scale patterns (Fig S10, SI). Specifically, both JRA-55 and MERRA-2 show that elevated Gulf SSTs are associated with lower atmospheric pressure over the Arabian Peninsula and higher pressure over Iran and the northern Arabian Sea, accompanied by a weakening of the Shamal winds, a strengthening of the monsoon flow over the southern and western Arabian Sea, and enhanced surface air temperature and total-column water vapor over the Gulf and the Arabian Peninsula (Fig. S10, SI). Similarly, analyses of summer atmospheric conditions associated with major climate variability modes (ENSO and NAO) were repeated using JRA-55 and MERRA-2 (Fig. S11 and Fig S12, SI).

The resulting patterns are consistent across all three reanalysis products. As with ERA5, La Nina (El Nino) conditions are associated with lower (higher) atmospheric pressure over the Arabian Peninsula (Fig S11, SI), while negative (positive) NAO phases correspond to lower (higher) pressure over the Arabian Peninsula and higher (lower) pressure over Iran and the northern Arabian Sea (Fig S12, SI). Overall, these comparisons indicate that the large-scale atmospheric circulation patterns associated with extreme summer SSTs in the Gulf are robust across multiple reanalysis datasets. Nonetheless, all three reanalyses share a relatively coarse spatial resolution, which may limit their ability to capture small-scale features such as the effects of complex orography. These limitations are further compounded by the scarcity of long, continuous meteorological observations in the region, which constrains the validation of reanalysis-based fields at local scales."

2) It should be noted that ENSO, IOD and NAO are not independent of each other. For instance, ENSO can affect IOD, IOD can affect ENSO, and NAO can also affect ENSO. The authors should examine the combined and relative effects of ENSO, IOD and NAO on SSTs in the Gulf using partial composite or regression analysis.

We agree with the reviewer that these climate modes are not independent but rather interact with one another. In the correlation matrix presented in the manuscript (Table 2, page 23), we show that ENSO is significantly correlated with both the IOD (r = 0.36, p < 0.01) and the ISM (r = 0.38, p < 0.01), whereas its correlation with the summer NAO is not statistically significant (r = 0.09, p > 0.05).

To further address this comment, we expanded the discussion of the potential interactions among these variability modes and their implications, citing key studies that have highlighted such linkages (e.g., Wallace and Gutzler, 1981; Kirtman and Shukla, 2000; Behera et al., 2006; Ashok and Saji, 2007; Folland et al., 2009; Cai et al., 2011; Jiménez-Esteve and Domeisen, 2018; Zhang et al., 2019; Xu et al., 2024) and relating them to our findings. Specifically, we added the following text to the revised manuscript (Section 3.5, lines 334–342):

"The climate variability modes considered in this study are not independent but can interact, resulting in cumulative or nonlinear effects. In particular, ENSO, IOD, and the ISM exhibit strong interdependence (Kirtman and Shukla, 2000; Behera et al., 2006; Ashok and Saji, 2007; Cai et al., 2011). Although their relationship is more complex, ENSO and NAO can also interact and produce synergistic effects (Wallace and Gutzler, 1981; Jimenez-Esteve and Domeisen, 2018; Xu et al., 2024), albeit much more weakly during summer (Zhang et al., 2019). Our cross-correlation analysis

confirms these links, showing that ENSO is significantly correlated with both the IOD (r = 0.36, p < 0.01) and the ISM (r = 0.38, p < 0.01), while its correlation with the summer NAO is not statistically significant (r = 0.09, p > 0.05) (Table 2). This weak summer ENSO-NAO coupling is consistent with Folland et al. (2009), who reported an asymmetric summer relationship, with La Nina episodes weakly associated with negative NAO phases, but no significant link between El Nino and positive NAO phases."

Second, following the reviewer's suggestion, we performed a partial regression analysis to isolate the unique relationship between Gulf summer SST and each of the four climate modes while controlling for the influence of the others (results shown in Fig. S8, SI). This analysis confirms that both the NAO and ENSO are significantly and negatively correlated with summer Gulf SSTs, whereas the IOD and ISM exhibit only weak and statistically insignificant correlations.

The partial regression analysis is first introduced in Section 2.6 of the Methods section (lines 172–176) as follows:

"Because these modes are not independent and exhibit mutual correlations, we further conducted a partial regression analysis to isolate the unique relationship between each mode and Gulf SST anomalies while statistically controlling for the influence of the others. This approach involves regressing the residuals of SST (after removing the effects of all other predictors) against the residuals of each individual climate mode (after removing the effects of the remaining modes)."

The results of this analysis are referenced in the Results section (Section 3.5, lines 342–346) of the revised manuscript:

"To isolate the unique relationship between Gulf summer SST and each of the four climate modes while controlling for the influence of the others, we conducted a partial regression analysis (Figure S8, SI). This analysis shows that both NAO and ENSO are significantly and negatively correlated with summer Gulf SSTs, whereas IOD and ISM display only weak and statistically insignificant correlations."

Finally, to quantify the relative importance of each predictor within the multiple regression framework, we applied a hierarchical partitioning approach (Mac Nally, 1996). This method is particularly suitable when predictors are correlated, as it is independent of the order in which variables are entered into the model. It decomposes the total explained variance into the unique contribution of each predictor and the portion jointly explained with other correlated predictors

(shared contribution). This analysis is first introduced in the Methods section (Section 2.6, lines 186–189):

"Finally, to partition the total explained variance into independent (unique) and joint (shared) components, we applied a hierarchical partitioning approach (Mac Nally, 1996), which systematically evaluates all possible combinations of predictors to quantify how much variance each explains individually versus in combination with others."

The results of this analysis are presented in Fig. S9 (SI) and discussed in the Results section (Section 3.5). This analysis confirms the dominant influence of the NAO and ENSO and shows that, owing to its weaker correlations with the other three modes, the NAO exhibits a higher unique contribution to the explained variance compared to the ENSO, which displays stronger associations with the IOD and ISM. A substantial fraction (approximately 20%) of the total explained variance is shared among all four modes. These findings are now included in the Results section (Section 3.5, lines 350–354):

"A decomposition of the total explained variance using hierarchical partitioning (Mac Nally, 1996) further confirms the dominance of NAO and ENSO, with the NAO showing a somewhat stronger unique contribution owing to its weaker correlations with the other three modes (Fig S9, SI). The analysis also reveals that approximately 20% of the total variance is jointly explained by all four modes, highlighting the intertwined nature of these large-scale climate drivers."

3) In addition to ENSO and the NAO, I would suggest that the authors also examine the role of Arctic sea ice anomalies in shaping extreme temperatures in the Gulf. Recent studies have indicated that Arctic sea ice anomalies could have a significant impact on ENSO, NAO and IOD. I suggest add some discussions. (https://doi.org/10.1175/JCLI-D-24-0419.1.; https://doi.org/10.1038/s41612-025-00936-x.; https://doi.org/10.1175/JCLI-D-23-0733.1.).

**Done.**

In response to the reviewer's suggestion, we examined potential links between summer Gulf SST and both Arctic sea ice and the Arctic Oscillation (Figs. 1 and 2). We found no statistically significant correlations between Gulf SST and either variable.

Fig 1. Zero-lag (left) and 6-month-lag (right) correlation between summer Gulf SSTs and Arctic sea ice area anomaly. Correlations are not statistically significant.

Fig 2. Zero-lag (left) and 6-month-lag (right) correlation between summer Gulf SSTs and the Arctic Oscillation index. Hatching indicates statistical significance at 95% confidence interval.

We conclude that, despite the potential connections between Arctic sea ice and the development of the IOD and ENSO highlighted in the studies cited by the reviewer, these links do not appear to play a critical role in determining Gulf summer SSTs. This may be because such relationships are weaker during summer, or because their influence does not substantially affect the summer atmospheric circulation over the Gulf region.

In the revised manuscript, we combined these two figures into a single figure presented in the Supplementary Information (Fig. S1) and mentioned this analysis in Section 2.5 (lines 142–144): "Other modes, such as the Arctic Oscillation, were also examined but showed very weak and statistically insignificant correlation with summer Gulf SST (Fig S1, Supplementary Information (SI)), and were therefore excluded from further analysis."

4) In addition to atmospheric heat fluxes, the tendency of SST should also be significantly impacted by oceanic dynamics, such as advection and upwelling/downwelling. The authors should analyse the SST tendency equation and investigate the role of these oceanic processes.

We agree with the referee that the SST tendency results from the balance between atmospheric heat fluxes and oceanic circulation and transport (mixing and advection).

To address this comment, we performed a complete heat budget analysis for the model surface layer during summer over the 39-year study period (Fig. 3 below). The SST tendency equation on which this analysis is based is now presented in a new subsection of the Methods section (Section 2.2, lines 95–100). The corresponding results are presented in Figure 2A of the revised manuscript (also shown below as Fig. 3).

Fig 3. Interannual anomalies of the temperature tendency term dSST/dt (black), atmospheric heat fluxes (red), and vertical (blue) and horizontal (green) heat transport fluxes, integrated over the entire Gulf. Correlation coefficients (r) in the legend indicate the Pearson correlation between each heat flux and the SST tendency term. Note that the tendency term and horizontal transport fluxes are plotted in °C per year, while the atmospheric heat fluxes and vertical transport fluxes are shown in °C per month, allowing all four terms to be displayed on the same scale.

As shown in this figure, SST tendency anomalies result from the near-compensation between anomalies in atmospheric heat fluxes (r = 0.88) and vertical heat transport (mixing and advection; r = -0.87), both of which are at least an order of magnitude larger than the SST tendency anomalies themselves and the anomalies in Gulf-integrated lateral heat transport (essentially equivalent to the heat transport from the Sea of Oman). Although vertical transport is comparable in magnitude to atmospheric heat fluxes, it is almost entirely driven by atmospheric heat flux anomalies (r = -0.99). Therefore, at the scale of the entire Gulf, oceanic transport (both vertical and horizontal) plays a much smaller role in SST variability. Indeed, as shown in Figures 2B and 2C of the revised manuscript, up to 92% of the variance in SST can be explained by local atmospheric heat fluxes in the northern Gulf (and 83% in the southern Gulf). The dominance of atmospheric heat fluxes over lateral heat transport likely reflects the Gulf's shallow (<30 m on average) and semi-enclosed nature (Strait of Hormuz width  $\approx$  42 km), which causes the contribution of local oceanic circulation to be strongly modulated by local atmospheric forcing.

In the revised manuscript, these new results are described in Section 3.1 (lines 217–225) as follows:

"A heat budget analysis of the surface layer over the study period reveals that anomalies in the SST tendency term dSST/dt primarily reflect the near-compensating effects of anomalies in atmospheric heat fluxes (r = 0.88\*) and vertical transport processes (mixing and advection; r = -0.87\*) (Fig. 2A). In contrast, anomalies in Gulf-integrated lateral heat transport—associated with heat exchange with the Sea of Oman—exhibit only a weak correlation with the SST tendency (r = 0.25; Fig. 2A). The strong anticorrelation between anomalies in atmospheric heat fluxes and vertical transport (r = -0.99\*) further indicates that vertical transport largely acts as a response to surface forcing rather than an independent driver of SST variability (Fig. 2A). Overall, at the scale of the entire Gulf, variations in atmospheric heat fluxes emerge as the dominant control on SST variability. Spatially, this influence is strongest in the northern Gulf, where atmospheric heat fluxes account for 92% of the SST variance (Fig. 2B), and remains dominant in the southern Gulf, accounting for about 83% (Fig. 2C)."

While oceanic transport (both vertical and horizontal) plays a relatively minor role in SST variability at the scale of the entire Gulf, its contribution can become more important locally, particularly near the Strait of Hormuz. However, since the present study focuses on the drivers of warm summer SSTs at the Gulf-wide scale, our emphasis remains on atmospheric heat fluxes, which account for the majority of the interannual variability. This caveat is explicitly noted in the revised manuscript (Section 5.2, *Caveats and Limitations*, lines 439–444):

"Finally, this study focuses on understanding extreme SSTs at the scale of the entire Gulf. Our large-scale analysis reveals that most of the interannual variability is driven by fluctuations in local atmospheric conditions, with only a limited contribution from remote oceanic influences. However, at more localized scales—particularly near the Strait of Hormuz—the influence of oceanic

circulation and heat transport from the Sea of Oman, and thus changes within the Sea of Oman itself, may become more significant and warrant further detailed investigation."

5) The authors discussed the possible influence of ENSO, NAO and IOD on SST anomalies in the Gulf. However, the underlying physical mechanisms have not been investigated in detail. The authors should examine the physical processes through which ENSO and NAO impact the formation of local atmospheric circulation. For example, Cheng et al. (2023; https://doi.org/10.1007/s00382-022-06616-3) indicated that AO/NAO-related atmospheric heating over the North Atlantic could trigger an atmospheric wave train from the North Atlantic to the northern Indian Ocean.

Understanding the mechanisms of global teleconnections that link large-scale climate variability modes to the modulation of regional summer atmospheric circulation over the Arabian Peninsula and Gulf region is an important open research question. However, a detailed investigation of these mechanisms is beyond the scope of the current study. This limitation is highlighted in the new section 4.2 "Caveats and Limitations," in the revised manuscript.

Nevertheless, the revised manuscript includes a brief discussion of mechanisms proposed in studies that have explored aspects of this question. For example, Cheng et al. (2023) proposed that atmospheric heating anomalies over the North Atlantic can trigger a Rossby wave train toward the northern Indian Ocean, potentially affecting the development of the Indian Ocean Dipole (IOD) in autumn. However, as our results show that the IOD has a limited impact on Gulf summer SST, likely due to the minor influence of Indian Ocean heat advection on interannual Gulf SST variability, this mechanism may contribute only marginally to summer Gulf SST variability.

Several studies have examined the connection between ENSO and summer atmospheric circulation in the region. For instance, Yu et al. (2015) identified a statistically significant relationship between ENSO and the timing of onset and termination of summer Shamal winds, which they linked to the Iranian summer heat low. However, that study did not propose mechanisms explaining the linkage between tropical Pacific SST anomalies and the development of the Iranian heat low. Attada et al. (2019) linked a strong Indian Summer Monsoon, generally favored during La Niña conditions, to increased adiabatic warming over the region due to enhanced subsidence in the middle and upper troposphere. This is consistent with our findings, which show increased subsidence in the upper troposphere over the Arabian Peninsula.

Finally, Kuman and Ouarda (2014) investigated the link between UAE winter precipitation and ENSO. They suggested that ENSO-driven changes in regional atmospheric circulation over the Arabian Peninsula are associated with shifts in planetary Rossby waves generated in the central Pacific due to SST anomalies, which then propagate into extratropical latitudes as cyclonic and anticyclonic features in the upper troposphere near the subtropical jet stream. While that study focused on winter conditions, similar mechanisms may contribute to ENSO-related changes in regional atmospheric circulation over the Arabian Peninsula during summer.

Regarding mechanisms linking NAO to regional atmospheric circulation, fewer studies have examined this relationship, particularly in summer. Chronis et al. (2011) showed that negative summer NAO is associated with lower cloudiness and higher temperatures over the eastern Mediterranean and the Anatolian Plateau. Folland et al. (2009) demonstrated that negative summer NAO is associated with lower pressure over the northern Arabian Peninsula and Iraq. Such weakening of the high-pressure system that typically dominates the eastern Mediterranean and western Arabian Peninsula in summer can reduce the pressure gradient across the Gulf and, consequently, decrease the frequency and intensity of Shamal winds.

To address the reviewer's comment, we added a synthesis of the above discussion to the revised manuscript in the new section 4.2 "Caveats and Limitations" (lines 427–438), as follows:

"While this study establishes robust statistical links between Gulf SST anomalies and major climate variability modes such as ENSO and the NAO, primarily through their associated modulation of surface pressure and wind patterns over the Arabian Peninsula and Iran, it does not explicitly examine the underlying physical mechanisms by which these remote modes influence regional atmospheric circulation. Understanding the dynamical pathways that connect Pacific and North Atlantic variability to the Arabian Peninsula's atmospheric conditions remains a complex problem that extends beyond the scope of the present analysis, which focuses on characterizing SST variability within the Gulf itself. Several previous studies have reported that ENSO and NAO modulate pressure systems over the Arabian Peninsula and Iran without fully explaining the mechanisms responsible for this modulation (e.g., Folland et al., 2009; Chronis et al., 2011; Yu et al., 2016). Other studies (e.g., Niranjan Kumar and Ouarda, 2014; Attada et al., 2019; Cheng et al., 2023) have proposed potential teleconnection pathways involving largescale Rossby wave trains or adjustments of the subtropical jet stream in the upper troposphere. Nevertheless, a comprehensive dynamical attribution has yet to be established. Future work combining observational analyses with targeted climate-model experiments will be required to elucidate these linkages in greater detail."

\*\*\*\*\*\*\*\*\*

**RC2: 'Comment on egusphere-2025-2948', Anonymous Referee #2, 03 Sep 2025**

The manuscript by Lachkar et al. addresses an important and timely topic by discussing the drivers of extreme summer sea surface temperatures in the Arabian Gulf. The authors combine an eddy-resolving ocean hindcast with ERA5 reanalysis, providing a framework to separate local and remote atmospheric influences. The work is well-motivated, as the Gulf is a critical hotspot of marine heat stress, with major ecological and socio-economic consequences. The manuscript is generally well-structured, the figures are clear, and the results are presented in a way that advances understanding of both local mechanisms (heat fluxes, Shamal winds) and large-scale teleconnections (ENSO, NAO). The finding that ENSO and NAO together explain over 50% of Gulf SST variability is particularly noteworthy and of interest for predictability. Overall, the paper makes a valuable contribution and is suitable for publication after some clarifications and refinements.

We sincerely thank the reviewer for their positive and encouraging feedback, and for the time and effort they invested in reviewing our manuscript.

**Specific Comments**

1) The current manuscript however distinguishes itself from earlier studies by focusing on summer SST extremes and their mechanistic links to atmospheric circulation, but the introduction could better highlight how this work fills the gap left by earlier Gulf studies that relied on multi-seasonal or coarser analyses. Also, how the Arabian gulf understanding can be important for precipitation extreme changes having said that arabian gulf as one of the moisture source for heavy rainfall (can be seen in Pathak et al. 2025).

Pathak, R., Dasari, H.P., Ashok, K. *et al.* Dynamics of intensification of extreme precipitation events over the Arabian Peninsula derived from CMIP6 simulations. *npj Clim Atmos Sci* 8, 126 (2025). https://doi.org/10.1038/s41612-025-01016-w

Following the referee's recommendation, we included the following statement highlighting the relevance of Gulf summer SST to extreme precipitation events in the region, given the role of the Gulf as a key source of atmospheric moisture (Lines 33-35 in the Introduction section of the revised manuscript):

"Finally, extreme summer temperatures in the Gulf may also affect regional extreme precipitation, given the Gulf's role as a key moisture source for the surrounding atmosphere (Pathak et al., 2025)."

We also now more clearly emphasize how the present study addresses gaps in earlier Gulf studies that relied on multi-seasonal data and coarser analyses. To this end, we added the following text to the revised manuscript (lines 47-55):

"However, most previous studies relied on coarse-resolution datasets (e.g., Purkis and Riegl, 2005), were spatially or temporally limited (e.g., Al-Rashidi et al., 2009; Nandkeolyar et al., 2013; Bordbar et al., 2024), or analyzed multi-seasonal data without isolating the summer period, when extreme SSTs predominantly occur (e.g., Nandkeolyar et al., 2013; Al Senafi, 2022; Bordbar et al., 2024). Moreover, no study has yet systematically investigated the mechanisms through which large-scale climate modes influence Gulf SSTs, nor quantified the relative contributions and interactions among these modes. As a result, the drivers of summer SST extremes and the mechanisms through which largescale climate modes influence Gulf SSTs remain poorly understood.

This study aims to fill these gaps by identifying the local and remote climatic drivers of extreme summer SSTs in the Gulf. Specifically, we address three key questions: (i) what are the dominant drivers of Gulf SSTs during summer?, (ii) what atmospheric conditions accompany extreme summer SSTs in the Gulf?, and (iii) to what extent, and through what mechanisms, are these conditions influenced by large-scale climate teleconnections?"

2) It is interesting to see that ENSO and NAO act largely independently and additively is interesting. Would be interesting to have a discussion expanded to compare with other ocean basins, where ENSO–NAO interactions are sometimes nonlinear.

While the interactions among different climate variability modes are complex and influenced by many factors, and thus extend beyond the scope of the current study, a brief discussion of this aspect has been added to the revised manuscript. Specifically, we expanded the discussion of potential interactions among the variability modes and their implications, citing key studies that have

highlighted such linkages (e.g., Wallace and Gutzler, 1981; Kirtman and Shukla, 2000; Behera et al., 2006; Ashok and Saji, 2007; Folland et al., 2009; Cai et al., 2011; Jiménez-Esteve and Domeisen, 2018; Zhang et al., 2019; Xu et al., 2024) and relating them to our findings. In particular, we cite Folland et al. (2009), who report a weak correlation between La Niña and negative summer NAO, but no significant correlation between positive NAO and El Niño.

The following text has been added to the revised manuscript (Section 3.5, lines 334–342):

"The climate variability modes considered in this study are not independent but can interact, resulting in cumulative or nonlinear effects. In particular, ENSO, IOD, and the ISM exhibit strong interdependence (Kirtman and Shukla, 2000; Behera et al., 2006; Ashok and Saji, 2007; Cai et al., 2011). Although their relationship is more complex, ENSO and NAO can also interact and produce synergistic effects (Wallace and Gutzler, 1981; Jimenez-Esteve and Domeisen, 2018; Xu et al., 2024), albeit much more weakly during summer (Zhang et al., 2019). Our cross-correlation analysis confirms these links, showing that ENSO is significantly correlated with both the IOD (r = 0.36, p < 0.01) and the ISM (r = 0.38, p < 0.01), while its correlation with the summer NAO is not statistically significant (r = 0.09, p > 0.05) (Table 2). This weak summer ENSO-NAO coupling is consistent with Folland et al. (2009), who reported an asymmetric summer relationship, with La Nina episodes weakly associated with negative NAO phases, but no significant link between El Nino and positive NAO phases."

Please also see our response to Reviewer #1's 2nd comment.

3) At LN170 and other places. The analysis finds a weak role for IOD and ISM. Given that the IOD is often invoked in regional SST variability, it would be helpful to provide a more explicit explanation of why its influence is muted in the Gulf, possibly due to the strong dominance of atmospheric fluxes over advective processes.

We thank the reviewer for this valuable observation. As noted by the referee—and discussed in our response to Reviewer 1's comment #4—interannual variability in summer Gulf SSTs is primarily driven by variability in atmospheric fluxes, with only a modest contribution from lateral ocean advection from the Arabian Sea. This limited influence is mainly due to the shallow and semi-enclosed nature of the Gulf, which restricts the propagation of temperature anomalies from the Arabian Sea into the Gulf, particularly during summer.

Following the reviewer's suggestion, we have revised the discussion of how ENSO, NAO, IOD, and ISM exert distinct influences on surface temperatures in the Gulf and the Arabian Sea, as shown in Fig. 14 (subsection 3.6: "Divergent Thermal Responses of the Gulf and Arabian Sea to Major Climate Modes"). In the revised manuscript, we specifically note that while several previous studies (e.g., Al-Rashidi et al., 2009; Nandkeolyar et al., 2013; Bordbar et al., 2024) have assumed that Gulf SSTs respond similarly to those of the adjacent Arabian Sea under large-scale teleconnections, our results reveal an anti-correlated behavior between the two basins. For instance, while positive IOD and El Niño events are typically associated with warming across the Arabian Sea, they coincide with weak-to-moderate cooling in the Gulf.

To explain this contrast, we explicitly propose in the revised manuscript two possible mechanisms underlying these divergent responses: (i) differences in how these modes modulate local winds—positive IOD and El Niño events are associated with weaker winds over the Arabian Sea but a modest strengthening of the Shamal winds over the Gulf (Fig. 10; Fig. S7, SI); and (ii) the stronger role of ocean circulation processes, including upwelling and lateral heat advection, in the Arabian Sea relative to the Gulf, owing to the shallow and semi-enclosed nature of the latter, which limits the propagation of temperature anomalies from the Arabian Sea into the Gulf, particularly during summer (Lachkar et al., 2024).

More concretely, in the revised Section 3.6 (lines 374–383), we now explain that:

"These contrasting responses between the Gulf and the Arabian Sea are noteworthy, as several previous studies (e.g., Al- Rashidi et al., 2009; Nandkeolyar et al., 2013; Bordbar et al., 2024) have assumed that Gulf SSTs respond similarly to those of the adjacent Arabian Sea under large-scale teleconnections. In contrast, our results reveal an anti-correlated behavior between the two basins. For instance, while positive IOD and El Nino events are typically linked to warming across the Arabian Sea, they coincide with weak to moderate cooling in the Gulf.

These contrasts likely arise from two main factors: (i) differences in how these modes modulate local winds—positive IOD and El Nino events are associated with weaker winds over the Arabian Sea but a modest strengthening of the Shamal winds over the Gulf (Fig. 10; Fig. S7, SI); and (ii) stronger role of ocean circulation including upwelling and lateral heat advection in the Arabian Sea relative to the Gulf, due to the shallow and semi-enclosed nature of the latter, which limits the propagation of temperature anomalies from the Arabian Sea into the Gulf, especially in summer (Lachkar et al., 2024)."

4) Please provide a schematic summarizing the coupled processes (weakening Shamal winds, enhanced subsidence, moisture build-up), and the same would be very useful for readers.

Such a schematic was already included in the original manuscript (Fig. 14), but it was previously referenced only in the Conclusion, which we acknowledge may not have been the most appropriate location. In the revised manuscript, we have moved it earlier (now presented as Fig. 7) and refer to it at the end of Section 3.2 in the Results section (line 269) to enhance its visibility and relevance.

5) Since as we know that ENSO and NAO are routinely monitored, please provide lead time with which Gulf summer SST anomalies could be anticipated.

Done.

As discussed in Section 4.3, there is potential for seasonal forecasting of summer Gulf SSTs with a 2–3-month lead time when initialized in late spring (e.g., May), particularly once the spring predictability barrier of ENSO (March–May) subsides and the skill of summer NAO prediction improves (Dunstone et al., 2023). In the revised manuscript, we clarify this point by adding the following text (Section 4.3, lines 455–457):

"Therefore, subseasonal-to-seasonal forecasts that incorporate North Atlantic and equatorial Pacific precursors may provide skill in predicting summer Gulf SSTs with 2-3 month lead time when initialized in late spring (e.g., May). This predictive skill is likely enhanced during strong ENSO and NAO phases, offering potential for early warning of summer MHWs in the region."

\*\*\*\*\*\*\*\*\*

RC3: Anonymous Referee #3, 30 Sep 2025

**General comments**

The paper investigates the dominant mechanisms driving extreme summer sea surface temperatures (SSTs) in the Arabian Gulf during 1980–2018, using regional hindcast simulations and ERA5 reanalysis data. The results show that extreme warming events are associated with pressure anomalies

that strengthen monsoonal winds in the western Arabian Sea while weakening local northwesterly winds (Shamal winds) over the Gulf. Enhanced monsoon circulation increases evaporation over the Arabian Sea, moisture transport into the Gulf, and promotes subsidence aloft, all of which trap heat near the surface. At the same time, weaker local winds reduce evaporative cooling, further amplifying warming. These processes are strongly modulated by large-scale climate variability, with ENSO and the NAO together explaining over 50% of the interannual SST variability. The warmest summers occur when La Niña and negative NAO phases coincide.

By examining both local and remote influences on the extreme summer conditions in the Arabian Gulf, the study provides valuable insights into the mechanisms behind regional extremes that can lead to marine heatwaves and broader ecological impacts. The work is well-motivated and methodologically thorough, particularly in its effort to disentangle the contributions of different drivers. I believe this paper will be of interest to the ocean extremes research community, and I recommend it for publication after minor revisions to improve clarity and presentation.

The authors are grateful to the reviewer for their constructive and encouraging comments, and for the time and effort devoted to evaluating our work.

I also have four more general comments:

1) Title precision: Since the study focuses on sea surface temperature, the title should be adjusted accordingly. I recommend: "Local and remote climatic drivers of extreme summer sea surface temperatures in the Arabian Gulf."

Thank you for this suggestion. We have revised the title as suggested.

2) Conclusion clarity: I strongly recommend revising the conclusion section to improve clarity and the overall flow of the narrative. Currently, it includes too much detail, which dilutes the impact of the key takeaways. The section would benefit from being more concise and focused, emphasizing the main messages rather than repeating specific points already covered earlier in the text.

Done. We have revised the Conclusion to remove unnecessary details and emphasize the key takeaways. Specifically, we deleted phrases such as "using both a model hindcast simulation and ERA5 reanalysis data," and "—typically associated with a strengthening of the Arabian heat low in

the lower troposphere and enhanced monsoon winds," as well as the references to specific years corresponding to La Niña or negative NAO events.

3) Use of acronyms: Acronyms should be defined at first use and then used consistently throughout. For example marine heatwaves (MHWs) are named in both the abstract and the introduction but the acronym is given only in the introduction. Please be coherent with them. Sometimes the sea surface temperature is written SST, other SSTs. Just decide for one and go with it through the whole paper.

Done. We have ensured a consistent use of acronyms, including MHW and SST, in the revised manuscript.

4) Study area map: A figure showing the geographical location of the study area would be very useful. Including key features such as the Strait of Hormuz and the Sea of Oman would help readers follow the discussion more easily in the results section.

Done. We have revised Figure 1B, which shows the study area, to include labels for the Gulf, the Sea of Oman, and the Strait of Hormuz, as suggested by the reviewer.

**Specific comments**

5) Line 6: 'Extreme summer temperature' is too general and could imply only air temperature while the study focuses on the sea surface temperature. Please modify accordingly.

Done. We have revised it to: "extreme summer SSTs in the Gulf" (line 6, abstract).

6) Lines 47-53 need to be improved.

First of all the aim is not so clearly reported. The authors go from the gap of knowledge to the research questions, making it difficult for the reader to understand the real objective of the work. Then, it is not so clear in line 47 to what previous studies the authors are referring to. Are those listed in the previous lines or are they new? If this is the case please cite them. I may guess from the sentence in line 47 that isolating summer SSTs in multi-seasonal data analyses could be helpful in identifying the drivers of extreme temperature in the Gulf and if that is the case, then it should be clearly stated followed likely by the aim of the work.

We are referring here to the studies cited in the previous paragraph. We have clarified this point in the revised manuscript and also improved the transition from the identified knowledge gap to the stated objectives of the paper. The revised text now reads:

"However, most previous studies relied on coarse-resolution datasets (e.g., Purkis and Riegl, 2005), were spatially or temporally limited (e.g., Al-Rashidi et al., 2009; Nandkeolyar et al., 2013; Bordbar et al., 2024), or analyzed multi-seasonal data without isolating the summer period, when extreme SSTs predominantly occur (e.g., Nandkeolyar et al., 2013; Al Senafi, 2022; Bordbar et al., 2024). Moreover, no study has yet systematically investigated the mechanisms through which large-scale climate modes influence Gulf SSTs, nor quantified the relative contributions and interactions among these modes. As a result, the drivers of summer SST extremes and the mechanisms through which largescale climate modes influence Gulf SSTs remain poorly understood.

This study aims to fill these gaps by identifying the local and remote climatic drivers of extreme summer SSTs in the Gulf. Specifically, we address three key questions:[...]"

7) Lines 55-63: information related to the results is irrelevant in the introduction section. It should be more appropriate for the discussion. It could be instead useful to describe how the paper is organised, for instance describing what information each section provides.

We respectfully disagree with the reviewer's comment. It is increasingly common, and considered good practice in many journals, to briefly summarize the main findings at the end of the introduction. This provides readers with a clear sense of the study's contribution and helps them follow the motivation for the subsequent analyses. We therefore prefer to retain this concise summary of key results.

Regarding the suggestion to include an overview of the manuscript structure, we feel that this is not essential as section overviews tend to repeat information that is already clear from the manuscript's organization and headings.

8) Line 67: Why is the local heat lux variability important to analyse in this work? The authors never mentioned that before and suddenly it appears here. It could be obvious the link between heat flux and extreme SST but it is not. Everything should be well clarified.

We have revised this to: "which is used to investigate the link between local atmospheric changes over the Gulf and large-scale atmospheric circulation." (lines 72–73, Section 2, *Methods*).

9) Line 122: At which hPa is the geopotential analysed? Is it considered for the whole column or the same as the wind vectors? Please specify.

As described in the text, the geopotential height as the specific humidity and wind vectors are analyzed at two levels: 850hpa and 300hpa (lines 132–133, Section 2.4, *Reanalysis Data*).

10) Line 154: What are the Gulf-wide SST anomalies? How do they differ from the general term SST anomalies in the Gulf? Or do they refer to the spatially averaged SSTs in the Gulf? Please clarify and write better keeping coherence throughout the whole text.

For more clarity, we have replaced "Gulf-wide SST anomalies" with "Gulf-mean SST anomalies" (line 171, Section 2.6, *Statistical Analysis*).

11) Line 158:Do you consider both the 90th and 98th percentiles and why? They could already represent a pretty extreme condition so is it like extreme and more extreme conditions you want to analyse? Please clarify and specify better.

In the revised manuscript, we limit the composite analysis to the 90th percentile. The most extreme cases (98th percentile) are very rare—occurring in only three summers when SST in more than one-third of the Gulf exceeded that level. These three summers are highlighted in Table 1 and discussed in the text. For more clarity, this statement has been revised to:

"we conduct a composite analysis, in which atmospheric fields are averaged during periods when Gulf SST anomalies exceed the 90th percentile, or when climate modes such as ENSO and NAO are in strong positive or negative phases. The most extreme summers are identified as summers where SST exceeds 2 standard deviations (98th percentile) over at least one-third of the Gulf." (lines 180-183, Section 2.6, *Statistical Analysis*).

**12) Line 167-170: It is not so clear to me what each neuron corresponds to.**

In the SOM analysis, each neuron represents a characteristic or prototype pattern learned from the input data and has the same dimensionality as the input. In this study, each neuron corresponds to a specific Gulf SST anomaly together with the concurrent intensity and phase of key climate variability

modes, represented by their respective indices. This explanation has been revised for greater clarity in the manuscript as follows:

"Each neuron (or map unit) represents a typical or prototype pattern learned from input observations and has the same dimensionality as the input. In this study, each neuron corresponds to a specific Gulf SST anomaly and the concurrent intensity and phase of key climate variability modes, represented by their respective indices." (lines 194-196, Section 2.7, *Self-Organizing Maps*).

13) Line 169-171: I am puzzled about this sentence in the middle of the description of a specific methodology: Based on stepwise regression results described in the previous section indicating that IOD and ISM do not significantly improve model performance, only ENSO and NAO are included in the SOM analyses. In which previous section have you described the results based on the stepwise regression? Is this maybe a distraction in the writing process? Because to me this sentence does not really make sense in this context.

We agree that this statement was misplaced. In the revised manuscript, the sentence has been removed from the Methods section and relocated to the Results section. It is now discussed after the presentation of the stepwise multiple linear regression analysis in Section 3.5, where we have added the following text:

"Since ENSO and NAO together explain most of the variance in Gulf SST, we applied the SOM analysis to three variables: Gulf SST as the dependent variable, and ENSO and NAO as the independent variables or predictors (Fig 13)." (Lines 356-358, Section 3.5, *Cumulative Impacts of Climate Variability Modes on Gulf Summer SSTs*).

14) Lines 200 and 205: The authors mention an air-sea temperature difference and the surface moisture gradient. It is okay to put the formula but it should be better to add this information and variables in the data and methods section so to have all clear what is going on in the workflow.

Done. This information is now more clearly presented in Section 2.3 ("Formulation of Surface Heat Fluxes") of the Methods, specifically in Equations (4) and (5). In the revised manuscript, we retain the formulas defining these gradients in the Results section but now explicitly refer to Equations (4) and (5) to better link them to the formulations of latent and sensible heat fluxes described earlier in the Methods

15) Figure 1, panel A: I recommend removing the yellow lines, as they may add confusion to an already color-dense plot. Moreover, the caption states that spatiotemporal variability (±1standard deviation) is represented by the grey shading, with no mention of yellow lines. This discrepancy may cause confusion for readers.

Done. The yellow lines have been removed as suggested.

16) Figure 2, panel A: Horizontal transport of the heat? How has this been calculated? It is not mentioned in the methods section and it should be there. Please add specific information in the methodology section. With regard to this and based on the caption of the figure, why has the heat transport been considered only from the Sea of Oman and not over the whole Gulf, where the anomalies of heat fluxes have been computed and analysed? This is not so clear to me. Panel B and C: I would suggest writing plainly Arabian Gulf instead of AG. That can be not so straightforward, especially when this is the very first time in showing this acronym.

In response to this comment, as well as to Reviewer 1's comment #4, we have added a detailed description of the full heat budget for the model's surface layer, based on the time evolution of SST (the SST tendency term), which now includes horizontal heat transport. This addition appears in the new Section 2.2, "SST Tendency Equation" (lines 95–100) of the revised manuscript.

Furthermore, we have revised Figure 2 to display the interannual anomalies of both the SST tendency terms and the associated heat fluxes (atmospheric, lateral, and vertical transport fluxes) shown in panel A (see also our response to Reviewer 1's comment #4).

We note that the lateral heat transport is integrated over the entire Gulf; however, due to the semi-enclosed nature of the basin, this is effectively equivalent to the heat exchange with the Sea of Oman. This clarification has been explicitly added to the caption of Figure 2 and to the text of Section 3.1 ("Summer SST Variability in the Gulf and Its Drivers," lines 217–225) in the revised manuscript, which now reads:

"A heat budget analysis of the surface layer over the study period reveals that anomalies in the SST tendency term dSST/dt primarily reflect the near-compensating effects of anomalies in atmospheric heat fluxes (r = 0.88) and vertical transport processes (mixing and advection; r = -0.87) (Fig. 2A). In

contrast, anomalies in Gulf-integrated lateral heat transport—associated with heat exchange with the Sea of Oman—exhibit only a weak correlation with the SST tendency (r=0.25; Fig. 2A). The strong anticorrelation between anomalies in atmospheric heat fluxes and vertical transport (r = -0.99) further indicates that vertical transport largely acts as a response to surface forcing rather than an independent driver of SST variability (Fig. 2A). Overall, at the scale of the entire Gulf, variations in atmospheric heat fluxes emerge as the dominant control on SST variability. Spatially, this influence is strongest in the northern Gulf, where atmospheric heat fluxes account for 92% of the SST variance (Fig. 2B), and remains dominant in the southern Gulf, accounting for about 83% (Fig. 2C)."

Finally, we have revised the labels in Panels B and C to read "Arabian Gulf" instead of "AG," as suggested by the reviewer.

17) Line 214: To what evaporative cooling refers? To the latent heat flux component or to another variable? It is not so clear when looking at the figures.

To clarify this point, we have added a sixth panel to Figure 4 (Fig. 4F), which shows the correlations between SST and evaporation. In the northern Gulf, evaporation (and thus evaporative cooling) is negatively correlated with SST, indicating that higher SSTs are associated with weaker-than-normal evaporation. This reduced evaporation reflects weaker winds during periods of extreme summer SSTs. In contrast, in the southern Gulf, SSTs and evaporation are positively correlated, owing to the enhanced surface humidity gradient and weaker winds. The addition of the new Fig. 4F, together with the revised text shown below, makes this statement clearer.

"In the southern Gulf and the Sea of Oman, where wind changes are weak or uncorrelated with SST anomalies, evaporation increases, leading to cooler SSTs (Fig. 4F). In contrast, in the northern Gulf, the pronounced weakening of the Shamal wind during warm SST summers offsets the modest increase in humidity gradient ( $\Delta Q$ ), resulting in reduced evaporation and, consequently, diminished evaporative cooling (Fig. 4F and Fig. S3, SI)." (Lines 241-244, section 3.1, *Summer SST Variability in the Gulf and Its Drivers*).

18) Figure 3: Composites are mentioned and explained, which is good but they should be described in the methodological section, not in the caption of a figure. Please adjust.

Done. We have retained this information in the figure caption for reference; however, as suggested by the reviewer, we now first describe it in Section 2.6 (Statistical Analysis) as part of the overview of the statistical and composite analyses. The added text in the revised manuscript reads:

"To quantify the typical SST response to variations in surface heat fluxes, we perform a composite of SST anomalies corresponding to the difference between high (> +1 SD above the mean) and low (< -1 SD below the mean) values of atmospheric heat fluxes." (Lines 177-179, section 2.6, *Statistical Analysis*).

19) Figure 4: There is air-sea humidity gradient as a new variable that has the same formula of the surface moisture gradient previously stated in lines 205. Please be coherent with the naming of formulas.

Done. In the revised manuscript, we now use the term "air—sea humidity gradient" consistently throughout.

20) Lines 214-215: The consistency of the results with the literature should be mentioned in the discussion section and not in the results one.

As this observation is not critical to the study, the statement has been deleted.

21) Line 225: Shamal winds are here defined as northwesterly winds but they have been already mentioned several times before. Please define them the first time you mention them.

**Done.**

We have moved the definition of the Shamal winds to their first mention in the Introduction. "This pressure configuration weakens the predominantly northwesterly winds over Iraq, most of the Arabian Peninsula, and the Gulf, known as Shamal," (lines 60-61, *Introduction*).

22) Line 232: In the described sentence, it is unclear whether the reference to Figure 6 applies to all panels (A–E) or only to panels A and E, which would be more logical since they are the only ones showing pressure systems.

Indeed the statement refers only to panels A and E. For more clarity, in the revised manuscript we have replaced (Fig. 6A–E) by (Fig.6A and Fig 6E) (line 259, section 3.2, *Influence of Large-Scale Atmospheric Circulation on Extreme Summer SSTs in the Gulf*).

23) Line 254-259: I suggest adding panel labels to Figure 8 and referring to them in the text, rather than repeatedly using "Figure 8." This would improve clarity and readability, while ensuring the panels are also clearly indicated in the figure itself.

**Done.**

Following the reviewer's suggestion, we have labeled the four panels (A, B, C, D) in Figure 8 of the original manuscript (now Figure 9 in the revised manuscript) and refer to them directly in the text to improve readability (lines 281-286).

24) Same in lines 277-280 for Figure 9 and in lines 284-289 for Figure 10.

Done. In the revised manuscript, we now refer to the individual panels in Figure 9 (now Figure 10) and Figure 10 (now Figure 11) in the text, as suggested (lines 304-316).

25) Line 263: What do you mean with 'part of the summer'? This is not so clear to me. Also the year 2007 is in the neutral phase of the NAO, not just 1996 and 1999.

We thank the reviewer for raising this unclear and potentially confusing statement.

We have revised the text to: "was either in a negative phase (1998, 2000, 2010, 2015, 2016, 2017), or in a neutral phase (1996, 1999, 2007) (Table 1)." (lines 289-290, section 3.3, *Impact of Major Climate Variability Modes on Extreme Summer SSTs in the Gulf*)

26) Line 266-267: The definition of extreme events as SSTs exceeding the 98th percentile should be included in the appropriate Methods section. In addition, I recommend clarifying in Table 1 that "ENSO" refers to El Niño in the negative phase and La Niña in the positive phase. This point was not entirely clear while reading.

**Done.**

In Section 2.6 (Statistical Analysis) of the Methods (lines 182–183), we now state:

"The most extreme summers are identified as those in which SST exceeds 2 standard deviations (98th percentile) over at least one-third of the Gulf."

Following the reviewer's suggestion, we have also added the following clarification to the caption of Table 1: "Positive and negative phases of ENSO correspond to El Niño and La Niña, respectively."

27) Section 4.1 and 4.2: These subsections still contain descriptions of results. The Discussion section should focus solely on interpretation, placing the findings in the context of the analyzed work, comparing them with existing literature, and highlighting both novelties and limitations. Limitations and/or caveats are in general missing in the discussion.

**Done.**

We have moved Sections 4.1 and 4.2 of the original manuscript to the Results section in the revised manuscript (now Sections 3.5 and 3.6).

Regarding the limitations/caveats comment, we have added a dedicated section titled "Caveats and Limitations" (new Section 4.2, lines 410–443), where we discuss the sensitivity of the results to the chosen reanalysis product and summarize the main limitations of the study. These include:

- 1) The relatively coarse resolution of the reanalysis datasets. Despite this, our results are robust across multiple products (ERA5, JRA55, and MERRA2), as demonstrated in our response to Reviewer 1's first comment.
- 2) The unexplored mechanisms through which large-scale teleconnections associated with ENSO and NAO influence local summer atmospheric circulation over the Gulf. While some processes are discussed based on relevant literature, a full investigation of these dynamical pathways is beyond the scope of the current study (see also our response to Reviewer 1's comment #5).
- 3) The study focuses on extreme SSTs at the scale of the entire Gulf. At more localized scales, particularly near the Strait of Hormuz, the influence of oceanic circulation and heat transport from the Sea of Oman may become more significant. Changes within the Sea of Oman itself could also play a role and warrant detailed investigation in future work.
- 28) Line 315: The SOM definition has already been described in the Methods section so there is no need to repeat it here.

As suggested, these details have been removed from this section, as they are already covered in the Methods section.

29) Line 342: The sentence: "These findings are consistent with previous studies of MHWs" is potentially misleading. It could imply that the study focused on marine heatwaves (MHWs), which is not the case. The study examines extreme summer SSTs, which alone do not necessarily indicate the occurrence of a marine heatwave. Since the section title already clarifies the main point of the discussion, I suggest rephrasing this sentence to avoid implying that MHWs were specifically investigated. Maybe something like: Our findings align with previous research on MHWs, indicating that the key drivers identified for extreme SSTs are consistent with those known to influence these prolonged warming events in the ocean.

Done. We have reformulated this statement as suggested to read:

"These findings align with previous research on MHWs, indicating that extreme SSTs often arise from a combination of local-scale processes and large-scale teleconnections" (lines 393–394, Section 4.1, *Gulf SST Extremes in the Context of Global MHW Drivers*).

---

## Author Response (AR2)

We thank the editor for the technical comment (shown below in red). Our response is provided beneath it, with the requested technical correction highlighted in blue in the revised manuscript (tracked-changes version).

**Detailed Comment**

Section 2.7. There are various expressions here, e.g. best matching unit, neuron weights, learning rate, that will not mean much to readers unfamiliar with SOM. Is Kohonen et al. (2000) a sufficient reference, or should another reference be given to cover such expressions?

**Response:**

To address this comment, we have added an additional reference (Kohonen, 2001), which provides a more detailed description of the SOM algorithm and the associated terminology. This new reference has been incorporated on page 7, lines 199 and 203 of the revised manuscript.

**The added reference is:**

Kohonen, T. (2001). *The Basic SOM*. In **Self-Organizing Maps** (Springer Series in Information Sciences, vol. 30). Springer, Berlin, Heidelberg. <a href="https://doi.org/10.1007/978-3-642-56927-2">https://doi.org/10.1007/978-3-642-56927-2</a> 3